# Transcriptomic-Driven Drug Repurposing Reveals SP600125 as a Promising Drug Candidate for the Treatment of Glial-Mesenchymal Transition in Glioblastoma

**DOI:** 10.3390/ijms26199772

**Published:** 2025-10-07

**Authors:** Kirill V. Odarenko, Marina A. Zenkova, Andrey V. Markov

**Affiliations:** Institute of Chemical Biology and Fundamental Medicine, Siberian Branch of the Russian Academy of Sciences, 630090 Novosibirsk, Russia; k.odarenko@yandex.ru (K.V.O.); marzen@niboch.nsc.ru (M.A.Z.)

**Keywords:** drug repurposing, drug repositioning, glial-mesenchymal transition, proneural-mesenchymal transition, glioblastoma, SP600125

## Abstract

Glioblastoma multiforme (GBM) is an aggressive brain cancer characterized by highly invasive growth driven by glial-mesenchymal transition (GMT). Given the urgent need for effective therapies targeting this process, we aimed to discover potential GMT inhibitors using transcriptomic-based repurposing applied to both approved and experimental drugs. Deep bioinformatic analysis of transcriptomic data from GBM patient tumors and GBM cell lines with mesenchymal phenotype using gene set variation analysis (GSVA), weighted gene co-expression network analysis (WGCNA), reconstruction of GMT-related gene association networks, gene set enrichment analysis (GSEA), and the search for correlation with transcriptomic profiles of known GMT markers, revealed a novel 31-gene GMT signature applicable as relevant input data for the connectivity map-based drug repurposing study. Using this gene signature, a number of small-molecule compounds were predicted as potent anti-GMT agents. Further ranking according to their blood–brain barrier permeability, as well as structural and transcriptomic similarities to known anti-GBM drugs, revealed SP600125, vemurafenib, FG-7142, dibenzoylmethane, and phensuximide as the most promising for GMT inhibition. In vitro validation showed that SP600125, which is most closely associated with GMT-related hub genes, effectively inhibited TGF-β1- and chemical hypoxia-induced GMT in U87 GBM cells by reducing morphological changes, migration, vasculogenic mimicry, and mesenchymal marker expression. These results clearly demonstrate the applicability of connectivity mapping as a powerful tool to accelerate the discovery of effective GMT-targeting therapies for GBM and significantly expand our understanding of the antitumor potential of SP600125.

## 1. Introduction

Glioblastoma multiforme (GBM) is the most common and aggressive brain tumor. The introduction of the Stupp protocol (maximal resection followed by radiotherapy with concurrent and adjuvant temozolomide) as the standard of care in 2005 improved median patient survival from 12.1 to only 14.6 months [1]. Despite the incorporation of additional therapeutic options, including nitrosourea-based alkylating agents, the anti-angiogenic antibody bevacizumab, and tumor-treating fields, into clinical management [1], the response rate in GBM remains exceedingly low. Even with a full course of therapy, only 4% of patients survive five years post-diagnosis due to high rates of GBM recurrence [2]. A primary driver of this is the highly invasive nature of GBM cells, which infiltrate healthy brain parenchyma, preventing complete surgical removal [3].

Bulk RNA sequencing (RNA-seq) data from The Cancer Genome Atlas (TCGA) initially categorized GBM tumors into four subtypes: classical, proneural, mesenchymal, and neural [4]. Subsequent studies, however, revealed that the neural subtype likely represents contamination by normal neural cells, leaving three principal GBM-specific subtypes [5]. Prognosis is most favorable for the proneural subtype and poorest for the mesenchymal subtype, which is associated with enhanced invasive potential and treatment resistance [5,6]. Key microenvironmental components, such as hypoxia, inflammation, and growth factors, can induce the transition of GBM cells to the mesenchymal subtype, a process known as glial-mesenchymal transition (GMT) [7].

GMT shares features with the epithelial–mesenchymal transition (EMT) observed in carcinomas, but it is an adaptation specific to the central nervous system. Like EMT, GMT causes GBM cells to adopt a mesenchymal-like phenotype, characterized by the upregulation of mesenchymal markers, such as Snail, N-cadherin, and vimentin, as well as increased cell motility and invasiveness [8]. However, GMT differs from EMT in several key aspects: (1) the absence of the canonical E-to-N cadherin switch [9], (2) hyperproduction of extracellular matrix components (e.g., collagens and fibronectin) [10], and (3) a more pronounced tendency of GBM cells to exhibit vasculogenic mimicry during mesenchymal transition compared to epithelial cells [11].

The combination of these traits with the lack of a basal membrane in GBM cells enables them to interact effectively with extracellular matrix components. This interaction allows the cells to migrate along these structures and spread diffusely into healthy tissue [12]. While GBM cells frequently display baseline mesenchymal characteristics, GMT is regarded as a process that substantially amplifies them [13]. Given the established link between GMT and the diffusive growth of GBM [12], coupled with its frequent activation in response to various stimuli (including growth factors, hypoxia, and temozolomide treatment) [13], targeting GMT has emerged as a promising strategy for GBM therapy [14,15,16].

Drug repurposing accelerates and reduces the cost of drug development by bypassing the pre-approval testing required for new drug candidates [17]. In GBM, this approach showed promising results. The combination of dabrafenib and trametinib, previously used to treat melanoma with the BRAF^V600E^ mutation, improved response rates in GBM patients with this mutation and was approved by the FDA in 2022 [18]. Regorafenib, a multikinase inhibitor used to treat gastrointestinal cancers, is currently in a phase 2/3 clinical trial (NCT03970447) and has demonstrated a survival benefit over lomustine in patients with recurrent GBM [19]. These findings demonstrate the effectiveness of drug repurposing in GBM, but further research is needed to make this therapy available to more patients.

Targeting GMT represents a novel approach for drug repurposing and is currently at the preclinical stage. Recent studies have shown that abemaciclib, a CDK4/6 inhibitor used in breast cancer treatment, and atomoxetine, prescribed for attention-deficit/hyperactivity disorder (ADHD), can inhibit GMT. In vitro, both drugs significantly reduce motility, invasion, stemness, and the expression of mesenchymal markers in GBM cell lines. In vivo, abemaciclib suppressed the growth of intracranial tumors, while atomoxetine enhanced the sensitivity of subcutaneous tumors to radiotherapy in mouse xenograft models [20,21].

The CLUE platform facilitates transcriptomic-based drug repurposing. It contains expression profiles derived from a variety of tumor cells exposed to approximately 3000 small-molecule compounds. By comparing these profiles with expression signatures of diseases, the platform finds potential drug candidates using a Connectivity Map method [22]. CLUE has identified potential therapeutic applications for thioridazine (an antipsychotic drug), repaglinide (an antidiabetic drug), and flubendazole (an anthelmintic drug) in GBM, and these findings have been successfully validated in preclinical models [23,24,25]. Although there are many reports describing the use of Connectivity Map-based searches for GBM-targeting drug candidates [25,26,27,28,29], to our knowledge, none of them have used GMT-related gene signatures as the search input. Given the efficacy of the Connectivity Map in identifying EMT inhibitors [30,31,32], this approach offers a promising avenue for the development of novel GMT-targeting anti-invasive drugs for GBM.

In this study, we employed a systematic approach to repurpose approved and investigational drugs to identify effective GMT inhibitors. We applied a comprehensive bioinformatics pipeline to analyze GMT-related genes and their networks, leading to the selection of a gene signature potentially involved in GMT regulation in both cell cultures and patient tumors. Using the identified GMT-related gene signature, we conducted a Connectivity Map analysis to identify small molecules with anti-GMT potency. Subsequent in silico screening based on multiple GMT-associated indicators identified SP600125 as the most potent anti-GMT drug candidate. In validation studies, SP600125 effectively suppressed GMT in two independent in vitro models, including transforming growth factor beta 1 (TGF-β1)-stimulated GBM cells and GBM cells with hypoxia induced by CoCl_2_. Thus, our findings demonstrate the validity of a transcriptomics-driven drug repurposing strategy for identifying novel GMT inhibitors. The obtained results provide a basis for developing compounds that target the highly invasive, mesenchymal-like phenotype of GBM cells.

## 2. Results

### 2.1. Exploring Glial-Mesenchymal Transition in TCGA-GBM Dataset

To approach GMT-targeting drug repurposing, a comprehensive bioinformatic analysis of publicly available GMT-related transcriptomic data was performed according to the study workflow depicted in Figure 1.

First, we examined GMT in patient-derived GBM tumors using the methodology previously described by Liang et al. [33]. Gene set variation analysis (GSVA) was used to evaluate the enrichment of Verhaak’s mesenchymal and proneural gene signatures in 168 patient tumors from the TCGA-GBM RNA sequencing (RNA-seq) dataset [34]. For each tumor, the GMT score was calculated by subtracting the proneural enrichment score from the corresponding mesenchymal enrichment score. As expected, the highest and the lowest GMT scores were found in mesenchymal and proneural tumors, respectively, while the classical subtype showed intermediate values (Figure 2A). The accuracy of GMT scoring is further confirmed by the low GMT score observed in the neural subtype (Figure 2A), which is now considered a mixture of tumor cells and normal neural tissue [35], and the association of high GMT score with worse patient survival in the TCGA-GBM cohort (Figure 2B).

To evaluate the accuracy of the GMT score in reflecting GMT, we divided GBM samples into “high GMT score” and “low GMT score” groups based on the median GMT score. Then, we examined their association with GMT-related processes in the tumor microenvironment. As expected, gene set enrichment analysis (GSEA) revealed that samples in the high GMT score group were enriched in epithelial–mesenchymal transition (EMT), hypoxia, inflammation, and tendency to activation of the TGF-β signaling pathway (Figure 2C). These processes accompany the acquisition of a mesenchymal-like phenotype by tumor cells [36]. Deconvolution of immune cells in GBM tissue revealed that the high GMT group, in contrast to the low GMT group, was characterized by a more heterogeneous immune landscape, enriched with a significantly larger set of immune cells, including dendritic cells, monocytes, macrophages, various types of T cells, B cells, mast cells, and granulocyte-monocyte and lymphoid progenitors (Figure 2D). In turn, the low GMT group was enriched with T helper 1 cells, CD4+ central memory T cells, plasma cells, eosinophils, and megakaryocyte-erythroid progenitors (Figure 2D). These findings suggest that tumors in the high GMT group are more hypoxic and immunologically active, which creates a favorable environment for GMT induction [37].

To better understand the genetic landscape governing GMT in 168 TCGA-GBM tumors, we performed weighted correlation network analysis (WGCNA) of their transcriptomes, which identified 21 modules of co-expressed genes, each denoted by a specific color (Figure 2E). For each tumor, the overall expression pattern of a specific co-expressed module was summarized by its eigengene, representing the first principal component of the expression data [38]. Subsequent module-trait relationship analysis revealed that the eigengene of the “blue” module had the strongest positive correlation with the GMT score (R = 0.73) (Figure 2F). This module, consisting of 694 genes, of which 100 genes overlapped with Verhaak’s mesenchymal signature, was selected for further analysis.

### 2.2. Identification of Hub Genes in the GMT Regulatory Network

To construct a robust gene signature for Connectivity Map analysis, we aimed to distinguish GMT-related transcriptomic changes in GBM cells from those in the tumor microenvironment. To achieve this, we first identified which genes in the “blue” module are activated in mesenchymal GBM cells, then analyzed their position within the GMT-associated gene network to select hub genes with strong regulatory potential (Figure 3A).

Analysis of the GSE192710 RNA-seq dataset identified 3718 differentially expressed genes (DEGs) upregulated in mesenchymal GBM cells compared to their proneural counterparts, with 160 of these genes located in the “blue” module (Figure 3B,C). The regulatory potential of selected 160 genes was further assessed by analyzing their position in the reconstructed GMT-associated gene network, using six key topological criteria (Degree, MCC, MNC, EPC, Closeness, and Radiality). This analysis revealed 31 hub genes that occupy central positions in the network, including signal transduction participants (*A2M*, *ADRB2*, *ARRB2*, *DAB2*, *JAK3*, *LYN*, *PIK3CD*, *STAT5A*), regulators of immune response (*C3*, *CSF1*, *CXCL8, IL10RA*, *IL15*, *IL18*, *IL4R*, *IL6R*, *IL7R*, *IL13RA1*, *THBD*, *TLR1*, *TLR2*, *TLR6*), cell death (*CASP10*, *CASP4*, *FAS*, *PRKCD*), and cell adhesion and invasion (*ITGAL*, *PLAUR*, *SELL*) (Figure 3D). Importantly, 25 of these hub genes are not included in Verhaak’s mesenchymal signature; therefore, we decided to investigate their association with GMT in more detail.

We further examined the correlation between hub gene expression and transcriptomic changes associated with GMT in TCGA-GBM tumors. As expected, the most hub genes were positively correlated with GSVA enrichment scores for EMT, inflammatory response, hypoxia, and TGF-β signaling, with the most promising associations shown for *CXCL8*, *IL4R*, *PLAUR*, and *THBD* (median R = 0.61, 0.62, 0.69, and 0.69, respectively) (Figure 4A). Furthermore, the majority of hub genes demonstrated the co-expression with mesenchymal markers (the most significant correlation revealed for *FAS*, *IL13RA1*, and *PLAUR* (median R = 0.53)) and a negative, albeit relatively weak, correlation with proneural markers (*JAK3*, *FAS*, and *PLAUR* showed the most pronounced R coefficients (median R = −0.42)) (Figure 4B). Interestingly, immune cell deconvolution revealed an association of hub genes mainly with the monocyte-macrophage lineage, with a predominance of pro-inflammatory M1 macrophages (median R = 0.67). Among other immune cells, a prominent association with hub genes was found for B cells, CD4+ effector memory T cells (CD4+ Tem), and mast cells (median R = 0.49, 0.49, and 0.38, respectively), all belonging to the high GMT score group (Figure 4C). The results obtained are consistent with published data on EMT. A gene cluster associated with monocytes, macrophages, and mast cells was previously observed after EMT induction in breast cancer cells [39], and CD4+ Tem were shown to be involved in regulating EMT of pneumocytes during fibrosis [40].

Thus, topological analysis revealed a regulatory core of the GMT, consisting of 31 hub genes associated with, and likely triggered by, inflammation and hypoxia in the GBM microenvironment.

### 2.3. Assessment of Clinical Relevance for GMT-Associated Hub Genes

To verify the association of hub genes with GMT in clinical settings, the Chinese Glioma Genome Atlas (CGGA) and Gravendeel datasets, containing transcriptomic and survival data from 218 and 155 GBM patients, respectively, were included in our study [41,42]. Similarly to the procedures applied to the TCGA-GBM data collection, we calculated GMT scores for each tumor in these datasets using GSVA-derived enrichments of Verhaak’s mesenchymal and proneural signatures (see Section 2.1 for details). Subsequent analysis revealed a strong positive correlation between hub gene expression and GMT scores across all clinical datasets (median R = 0.60, 0.63, and 0.55 for TCGA, CCGA, and Gravendeel, respectively), similar to the correlation profile of mesenchymal markers (median R = 0.64, 0.60, and 0.61, respectively) and opposite to that of proneural markers (Figure 5A). In addition, hub gene expression exhibited a similar, albeit less robust, correlation with GSVA-derived enrichment of the EMT hallmark signature in tumors, with the most pronounced coefficients observed in the TCGA and CCGA datasets (median R = 0.51 and 0.50, respectively) (Figure 5A). The obtained data independently confirmed the relevance of hub genes as GMT participants and novel promising markers of mesenchymal-like phenotype in GBM patients.

To determine the extent to which hub genes have been studied in the context of GMT, we next performed text mining analysis. Since “GMT” is a relatively new term, we analyzed the co-occurrence of hub gene names with more established terms such as “GBM”, “EMT”, “metastasis”, and “invasion” in PubMed abstracts. As illustrated in Figure 5B, *CASP10*, *IL10RA*, and *IL13RA1* did not co-occur with any of these terms and can therefore be considered as novel candidate markers for GBM malignancy. The remaining genes have already been described in scientific texts in the context of “invasion” and “metastasis”, and co-occurrence with more specific terms, such as EMT (*DAB2* and *THBD*), GBM (*IL4R* and *TLR2*), or EMT and GBM together (13 genes) was shown for a number of them (Figure 5B), confirming the validity of our in silico analysis.

Consistent with this, a Cox univariate analysis of the TCGA-GBM, CGGA, and Gravendeel datasets revealed a significant association between survival outcomes in patients with GBM and 24 hub genes. Among them, *IL4R*, *A2M*, *FAS*, and *TLR2* showed this association in two transcriptomic sources, while *CASP4*, *IL13RA1*, *PLAUR*, and *THBD* were associated with poor prognosis in all three datasets (Figure 6A). Further stratification of patients by median expression levels of these genes and Kaplan–Meier analysis revealed worse survival in patients with higher expression of *CASP4*, *PLAUR,* or *THBD* (detected in two of the three datasets analyzed) (Figure 6B–D). In the case of *IL13RA1*, its high expression was associated with poor prognosis only in the TCGA-GBM dataset (Figure 6B).

Collectively, consistent results from three independent clinical datasets confirmed the association of revealed hub genes with GMT and GMT-related processes in GBM patients, underscoring their potential as a promising gene signature for transcripromics-based drug repurposing to block GMT.

### 2.4. Drug Repurposing via Connectivity Mapping

The revealed GMT-associated hub gene signature was uploaded to the CLUE web tool for Connectivity Map analysis, identifying 120 small-molecule compounds that reduce the expression of these genes in various human cancer cell lines independently of cell origin (Appendix A). As shown in Figure 7A and Appendix A, the largest group of identified drugs comprised 35 neurotransmitter agents, including modulators of adrenergic receptors (7 compounds), acetylcholine system (6), and serotonine system (6). The second largest category consisted of 25 compounds that target kinases, including VEGFR (4), JNK (2), GSK-3 (2), ERK5 (2), and LRRK2 (2). The third category included 12 inhibitors of non-kinase enzymes, primarily focusing on methyltransferases (3), peptidases (3), deacetylases (2), and prenyltransferases (2). Beyond these major classes, we also identified several additional drug categories, namely modulators of apoptosis (10), nuclear receptors (9), and membrane transport (5), antimicrobials (7), anti-inflammatory agents (5), and others (12). Notably, the Connectivity Map results included temozolomide, an FDA-approved GBM drug, and vemurafenib, which has shown durable anti-GBM activity in patients with the BRAF^V600E^ mutation [1].

Since the blood–brain barrier (BBB) presents a major obstacle in GBM drug development [43], the BBB permeability of the identified drugs was further assessed using two independent chemoinformatics tools, AlzPlatform and PreADMET. Temozolomide and doxorubicin served as reference compounds [44] and showed high and low BBB permeability, respectively, confirming the validity of the data obtained (Figure 7B). Results from both platforms indicated that 59 out of 120 compounds had comparable or higher BBB permeability scores than temozolomide, and these compounds were selected for further study (Figure 7B, Appendix A).

Next, a text mining analysis of PubMed Central articles was performed to determine whether the identified molecules had been studied in the context of cancer malignancy. As shown in Figure 7C, 22 out of 59 compounds were associated with the terms “metastasis” and “invasion”, of which 15 drugs were mentioned in abstracts related to GBM (naloxone, navitoclax, and phenothiazine), EMT (estrone, Lrrk2-IN-1, PCI-34051, and physostigmine) or both these terms (aspirin, estradiol, GSK-3 inhibitor II, L-BSO, resveratrol, SP600125, temozolomide, and vemurafenib). Further manual text analysis revealed that the listed EMT-associated drugs can be divided into three groups according to their effect on EMT: EMT inhibitors (aspirin [45,46], resveratrol [47,48], SP600125 [49,50], Lrrk2-IN-1 [51], PCI-34051 [52]), EMT inducers (estradiol [53,54], estrone [55], L-BSO [56,57], physostigmine [58]), and context-dependent EMT modulators (GSK-3 inhibitor II [59,60], temozolomide [61,62,63,64,65], and vemurafenib [66,67,68,69]). Similar effects of some of these compounds on GMT have also been reported previously, namely, suppression of this process by aspirin (basal GMT status) [70], resveratrol (TGF-β-induced GMT) [71] and SP600125 (IL33-induced GMT) [72] and its activation by estradiol [73]. Thus, these data confirm the validity of our transcriptome-based search for potent GMT modulators and the expediency of further analysis.

Given the crucial negative role of P-gp in the regulation of the efficiency of xenobiotic accumulation in brain tissue [74], we next asked whether the identified hit compounds were substrates for this efflux pump protein. To address this issue, the top 12 molecules with the highest BBB permeability scores (Figure 7B) were uploaded to three chemoinformatics platforms, LiverTox Workspace, SwissADME, and ADMETlab, and their substrate specificity for P-gp was predicted. As shown in Figure 7D, SP600125, dibenzoylmethane, FG-7142, vemurafenib, and phensuximide were recognized as non-substrates for P-gp by all in silico tools, confirming their promising application for GBM treatment.

Then, we assessed the similarity of the identified non-P-gp substrate compounds to GBM drugs (both FDA-approved and investigational) in terms of chemical structure and transcriptomic effects. As shown in Figure 8A,B, all of the selected hits were characterized by low or moderate structural similarity with GBM drugs (Tanimoto coefficients of less than 0.7). However, four out of five of the selected hits exhibited similar transcriptomic changes to those of temozolomide. Furthermore, vemurafenib and SP600125 exhibited high connectivity scores with procarbazine (a DNA alkylating agent), irinotecan (a topoisomerase I inhibitor), and cediranib (a VEGFR inhibitor) (Figure 8B). Dibenzoylmethane elicited a distinct transcriptomic profile compared to all GBM drugs (Figure 8A), which can be explained by its unique structural features that differ significantly from those of the analyzed GBM drugs (Figure 8B).

Thus, multiparametric in silico analysis revealed 22 BBB-permeable drugs capable of being repurposed for GMT treatment, among which SP600125, dibenzoylmethane, FG-7142, vemurafenib, and phensuximide were characterized by the most advantageous drug-like properties.

### 2.5. Effects of SP600125 on GMT Induced by TGF-β and Hypoxia

To select a compound for testing GMT inhibitory activity in vitro, we ranked hit compounds according to the strength of interactions with the GMT-related 31 hub gene signature. This involved (a) retrieving primary protein targets of hit compounds from the Search Tool for Interactions of Chemicals (STITCH) database and (b) mapping their interactions with hub gene signature using the Search Tool for the Retrieval of Interacting Genes/Genomes (STRING), resulting in a chemical-protein-gene network (Figure 9A). As shown in Figure 9A, only vemurafenib, SP600125, and FG-7142 formed edges with GMT-associated hub genes through their primary targets. Dibenzoylmethane and phensuximide had no edges with the reconstructed network at all, apparently affecting the GMT-related hub gene signature indirectly (Figure 9A). Since SP600125 had the highest centrality score among the analyzed hits (Figure 9B) and demonstrated a higher BBB permeability score than vemurafenib and FG-7142 (Figure 7B), this compound was finally selected for in vitro verification testing.

SP600125 is a multitarget compound that inhibits not only JNK1/2 (MAPK8/9) but also several other proteins, including six identified in the STITCH database (Figure 9A), as well as the recently reported AhR [75], AURKA [76], and MELK [77]. To determine which of these targets mediates the anti-GMT potential of SP600125, we assessed the effects of other JNK inhibitors and the knockdown of SP600125’s protein targets on the GMT signature. As shown in Figure 9C, only the JNK inhibitor CC-930 (tanzisertib) produced a negative connectivity score similar to SP600125 (Appendix A), while other inhibitors yielded positive scores. These results suggest that the anti-GMT potential of SP600125 is determined not solely by JNK1/2 inhibition but also by its effects on other targets. Indeed, analysis of knockdowns revealed that the inhibition of MELK and FLT3 may also underlie the GMT targeting effect of SP600125 (Figure 9C, Appendix A). Furthermore, JNK1 (MAPK8), but not JNK2 (MAPK9), was found to be associated with the GMT signature (connectivity scores of -90.8 and 10.86, respectively) (Figure 9C). Thus, our findings demonstrate that the effect of SP600125 on GMT is mediated by its inhibitory action on JNK1, MELK, and FLT3, which requires further experimental validation.

The effects of SP600125 on the viability of human GBM cells (U87 and U118), mouse GBM cells (EPNT-5 and GL261), and human fibroblast hFF3 cells were evaluated using the MTT assay after 24 and 48 h. As shown in Figure 10A, SP600125 reduced the viability of all GBM lines in a dose and time-dependent manner. Among the cell lines tested, U87 cells were the most sensitive to SP600125 (IC_50_^24h^ = 19.6 ± 2.5 μM, IC_50_^48h^ = 7.3 ± 0.9 μM). At 24 h, EPNT5 showed the least sensitivity (IC_50_^EPNT5^ = 104.7 ± 29.2 μM), while U118 and GL261 were more responsive (IC_50_^U118^ = 58 ± 6.8 and IC_50_^GL261^ = 65.2 ± 18.5). However, by 48 h, the effect was similar among these three lines, with IC_50_ values around 40 μM. Notably, cytotoxicity against non-malignant hFF3 cells was significantly lower than in GBM cells (IC_50_^24h^ = >100 μM, IC_50_^48h^ = 87.1 ± 22.7 μM) (Figure 10A).

Previous studies have explored the impact of SP600125 on mesenchymal transition, utilizing two different incubation strategies, namely long-term exposure [50,72,78] and short preincubation [49,79,80]. To evaluate how these approaches affect U87 cell motility, a key GMT-related process, we performed a wound healing assay. Long-term incubation with non-toxic concentrations of SP600125 modestly decreased U87 cell migration potential, with the highest dose (2.5 μM) reducing motility by 19.3% (24 h) and 21.3% (48 h) compared to the control. In contrast, a brief 30 min pretreatment of cells with 20 μM SP600125 had a significantly more pronounced effect, suppressing cell motility by 1.5-fold and 2.1-fold after incubation without the compound for 24 and 48 h, respectively (Figure 10B and Appendix A). The effect of the pretreatment regimen on cell motility was verified in U118 and GL261 cells (Figure 10C and Appendix A). Based on these findings, we selected a short preincubation scheme to assess the effects of SP600125 on GMT.

Since our repurposing strategy for SP600125 was based on the GMT gene signature, which positively correlates with TGF-β signaling and hypoxia (Figure 4A), these two factors were selected as GMT inducers for the verification study. To the best of our knowledge, the anti-GMT potency of SP600125 has been previously studied only in the immune-related model of IL33-induced GMT [72]. However, its ability to prevent GBM cells from acquiring mesenchymal-like characteristics under TGF-β and hypoxic conditions had not been explored.

It was shown that TGF-β1 stimulation at 50 ng/mL for 48 h significantly altered U87 cell morphology, increasing elongation (aspect ratio (AR)) by 37.5% and cell area by 74.4% compared to control. Preincubation with 20 μM SP600125 for 30 min before TGF-β1 stimulation partially reversed these changes, reducing cell area by 26.2% without affecting AR (Figure 11A). This partial mitigation of TGF-β1-induced morphological changes was verified in U118 cells. Interestingly, TGF-β1 increased elongation but decreased the size of U118 cells, and SP600125 reduced both these parameters (Appendix A).

Consistently, sequential incubation of U87 cells with SP600125 and TGF-β1 significantly affects their transwell migration. As shown in Figure 11B, TGF-β significantly enhanced cell penetration through the porous bottom of the CIM-Plate, increasing the normalized cell index (ΔNCI) by 8.7 compared to the control at 24 h, whereas pretreatment with SP600125 reduced this parameter 2.3-fold (ΔNCI = 3.8), exerting a suppressive effect throughout the experiment. Notably, in the absence of TGF-β1, SP600125 had no effect on cell migration (Figure 11B).

The GMT was known to induce vasculogenic mimicry, a process where tumor cells establish structures that resemble new blood vessels [81,82]. Microscopic examination revealed that U87 cells spontaneously formed closed tube-like structures, creating a mesh. Following a 48 h induction with TGF-β1, the mean size of these meshes increased by 35.6%. In contrast, preincubation with SP600125 significantly suppressed the formation of the vascular network, resulting in a 2.7-fold reduction in mesh size and the prevalence of branches in the network structure (Figure 11C).

Next, we conducted RT-qPCR to assess markers of the mesenchymal state in U87 cells. TGF-β1 treatment increased the expression of fibronectin (*FN1*), N-cadherin (*CDH2*), Slug (*SNAI2*), and uPAR (*PLAUR*) by 2.8-, 14.4-, 1.9-, and 2.2-fold, respectively. In contrast, preincubation with SP600125 reduced these levels to near control values (Figure 11D). To confirm these changes at the protein level, Western blot and immunofluorescence analyses were performed. Although TGF-β1 induced *CDH2* mRNA expression (Figure 11D), it did not increase N-cadherin protein levels at 48 h (Figure 11E). SP600125 suppressed N-cadherin expression in TGF-β1-treated U87 cells, lowering it below the control value (Figure 11E and Appendix A). TGF-β1 also significantly enhanced vimentin staining in U87 cells, which was reduced by SP600125 (Figure 11F). These findings show that SP600125 attenuated the phenotypic shift toward the mesenchymal state.

We demonstrated that the GMT signature is associated with an increased presence of immune cells within tumors (Figure 4C). GBM is known to shape its microenvironment by secreting immune molecules such as IL-6, IL-1β, and TGF-β1 [83]. Therefore, we assessed whether SP600125 modulates the immunoregulatory activity of U87 cells undergoing GMT. As shown in Figure 11G, TGF-β1 treatment increased IL-6 and TGF-β1 expression by 1.5- and 2.5-fold, respectively, but decreased IL-1β expression by 2.8-fold. Interestingly, SP600125 reversed all TGF-β1-mediated effects: IL-1β mRNA levels were restored to 59% of control, TGF-β1 expression was reduced by 1.6-fold, and IL-6 levels dropped to 57% of control (Figure 11G).

Finally, the pharmacological potency of SP600125 was evaluated in U87 cells subjected to CoCl_2_-induced chemical hypoxia. As shown in Figure 12A, CoCl_2_ significantly altered the morphological characteristics of GBM cells. However, unlike TGF-β1, its effect was more pronounced on AR (a 54.8% increase over control) than on cell area (a 33.9% increase) (for TGF-β1, these parameters were 37.5% and 74.4%, respectively (Figure 11A)). Preincubation of CoCl_2_-treated cells with SP600125 for 30 min decreases AR by 20.2% without affecting the cell area (Figure 12A).

In addition to altering cell morphology, CoCl_2_ increased vasculogenic mimicry formation in U87 cells by 38% compared to the control. SP600125 decreased this parameter by 2.2-fold (Figure 12B). The CoCl_2_-induced acquisition of a mesenchymal phenotype was accompanied by the upregulation of key mesenchymal markers and GMT-related hub genes (Figure 12C). Compared to the effect of TGF-β1 (Figure 12D), these changes were less pronounced but still statistically significant. Specifically, CoCl_2_ increased the expression of N-cadherin, Slug, IL-1β, and uPAR by 1.4-, 1.8-, 1.3-, and 1.2-fold, respectively, but had no effect on fibronectin. SP600125 counteracted the CoCl_2_-induced upregulation of N-cadherin, IL-1β, and uPAR, returning their expression to control levels. Surprisingly, SP600125 did not affect Slug expression in this model (Figure 12C).

Consistent with the strong correlation between the GMT signature and the pro-inflammatory M1-like macrophage gene profile (Figure 4C), our analysis showed that CoCl_2_ treatment increased IL-6 and IL-1β mRNA levels by 2.7- and 1.3-fold, respectively (Figure 11D). SP600125 reduced IL-1β to the control level, and, unexpectedly, further enhanced the CoCl_2_-induced increase in IL-6 mRNA by 1.7-fold (Figure 11D). To investigate the latter effect in greater detail, we assessed the influence of SP600125 on IL-6 at the protein level in CoCl_2_-treated cells. In line with the mRNA data, CoCl_2_ increased IL-6 secretion by 7.3-fold in U87 cells (Figure 11E), whereas SP600125 exerted a slight inhibitory effect, reducing IL-6 secretion by 12%. Thus, despite some discrepancy between the RT-PCR and ELISA results, it was demonstrated that hypoxia-induced GMT is indeed associated with upregulation of pro-inflammatory cytokines, and that SP600125 partly suppresses this effect. In addition, CoCl_2_ was found to increase TGF-β1 mRNA by 2.1-fold compared to the control, and SP600125 did not affect this upregulation (Figure 11D).

Thus, in vitro experiments clearly confirmed the potent GMT inhibitory activity of SP600125, which supports the results of our in silico analysis. Minor differences were observed in GMT induction by TGF-β1 and CoCl_2_, but SP600125 significantly prevented GBM cells from acquiring key mesenchymal-like features in both models, including morphological changes, vasculogenic mimicry, motility, and the expression of mesenchymal markers.

Taken together, our findings (a) demonstrate the promise of using transcriptomic data for anti-GMT drug repurposing, (b) significantly expand insights into the anti-GBM potential of SP600125, and (c) provide valuable information on hub genes sensitive to GMT inducers. The protein products of the identified hub genes could serve as diagnostic biomarkers and novel potential therapeutic targets for managing GBM malignancy.

## 3. Discussion

Recent research shows that tumor microenvironment factors, such as hypoxia and inflammation, drive GBM cells from a proneural to a mesenchymal subtype, promoting invasive growth and reducing treatment efficacy [37]. Given the strong association of GMT with glioma progression and the poor prognosis of GBM patients, GMT blockers could serve as a promising new class of anti-glioblastoma drugs. Despite the successful use of omics-based drug repurposing to find novel inhibitors of GBM cell viability [84], to our knowledge, this approach has not yet been applied to the search for anti-GMT compounds, highlighting the high novelty of the present study.

To date, several key intracellular factors controlling GMT have been identified (as reviewed in [12,37,85]), yet the subtle regulatory mechanisms remain incompletely understood. Our transcriptomic analysis provides new insights into this framework. We identified a set of GMT-susceptible genes that occupy hub positions in the GMT-associated gene network and exhibit transcriptional behavior similar to known mesenchymal markers (Figure 4B). These characteristics enabled the use of this gene set not only as a signature for identifying anti-GMT compounds but also to uncover additional features. Specifically, a significant fraction of these genes is associated with immune response (Figure 4A). xCell deconvolution analysis revealed that their expression is linked to monocyte/macrophage lineage signature, more strongly with the pro-inflammatory M1 type than with the tumor-associated M2 type (Figure 4C). This finding is consistent with published data. Hailong et al. demonstrated that the progression of high-grade glioma is specifically associated with the pro-inflammatory phenotype of brain-resident macrophages [86]. Furthermore, neuroinflammation has been shown to promote GBM malignancy [87]. Notably, *IL13RA1*, *IL4R*, and *IL10RA*, identified as GMT-related hub genes in our analysis (Figure 4A), have been previously included in immune-related gene signatures associated with the mesenchymal GBM phenotype [88] and poor patient prognosis [89].

Among the identified hub genes, *CASP4*, *IL13RA1*, and *THBD* are of particular interest due to their nodal location in the GMT-related regulome (Figure 3D) and negative association with GBM patient survival (Figure 6B–D). Although *CASP4* and *THBD* are part of Verhaak’s mesenchymal signature (Figure 6A), and *IL13RA1* is linked to poor survival in GBM patients [88], their direct role in GMT remains unclear. According to published data, these genes play an important role in regulating the malignant characteristics of GBM cells. Caspase 4 (*CASP4*) is more abundant in GBM than in normal brain tissue [90], and its activation induces ER stress-associated GBM cell death, highlighting its therapeutic significance [91,92,93,94]. Interleukin 13 receptor α1 (*IL13RA1*) is a marker of GBM progression and survival [95,96], while thrombomodulin (*THBD*) regulates GBM chemoresistance [97]. These genes have been implicated in processes related to acquiring a mesenchymal-like phenotype. For example, CASP4 regulates actin assembly and focal adhesion dynamics in GBM cells, thereby enhancing their migration [98,99,100]; IL13Rα1 induces key EMT-related transcription factors ZEB1 and β-catenin in colon and gastric cancer cells [101,102]; and THBD inhibits EMT in colon, bladder, and lung cancer cells [103,104,105]. Thus, our bioinformatics analysis had not only revealed a GMT-related hub gene signature applicable for drug repurposing but also identified novel GMT-sensitive targets for therapeutic control of GBM invasive growth.

Computational methods accelerate drug repurposing. For example, machine learning efficiently extracts key patterns from large multiparametric datasets to build prognostic models [106,107]. Using such models, anti-GBM activities were found for vortioxetine (an antidepressant), cetirizine (an antihistamine), and lovastatin (a lipid-lowering drug) [108,109,110]. The Connectivity Map, developed by the Broad Institute, identifies biologically active small compounds by comparing their transcriptomic responses in tumor cells with disease gene signatures [22]. This method led to the discovery of anti-GBM activity in thioridazine (an antipsychotic drug), repaglinide (an antidiabetic drug), and flubendazole (an anthelmintic drug) [23,24,25].

In our study, we used the Connectivity Map to identify small molecules with negative connectivity to the GMT signature, indicating their anti-GMT potency (Figure 7A). Given the extensive list of predicted compounds, which included diverse agent classes (Figure 7A), we applied four key selection criteria to identify hit compounds: (i) BBB permeability (Figure 7B), (ii) lack of P-gp substrate specificity (Figure 7D), (iii) reported activity in GBM or EMT studies (Figure 7C), and (iv) similarity in chemical structures or transcriptomic effects to known GBM drugs (Figure 8). This filtering identified SP600125, vemurafenib, FG-7142, dibenzoylmethane, and phensuximide as promising novel GMT inhibitors (Figure 7D). SP600125 was selected for further in vitro validation (Figure 10, Figure 11 and Figure 12) based on its superior connectivity within the GMT-associated network (Figure 9A, B) and high BBB permeability score (Figure 7B). The subsequent successful confirmation of its anti-GMT activity (Figure 11 and Figure 12) not only validates our computational strategy but also underscores the potential of the other candidates as promising therapeutics for targeting GMT.

Vemurafenib: This BRAF inhibitor suppresses EMT in colorectal cancer [67] and melanoma [68] cells and demonstrates prolonged disease control in clinical trials for high-grade glioma [111]. However, since resistance to vemurafenib in melanoma led to a re-emergence of the mesenchymal phenotype [69], future evaluation of its anti-GMT potential should consider the chemoresistant status of GBM cells.Dibenzoylmethane: This compound inhibits the viability of GBM cells at micromolar concentrations [112] and blocks the invasion of breast cancer cells induced by phorbol 12-myristate 13-acetate [113], a known EMT inducer [114].FG-7142: Although there are no reports on the direct effect of this GABAA receptor agonist on GBM cells, its anti-GMT potency warrants investigation. Activation of the GABAA receptor suppresses brain cancer [115]. Furthermore, moxidectin, an analog of FG-7142, significantly inhibited cancer stem cell properties of medulloblastoma cells [116], which are closely associated with the mesenchymal phenotype of CNS tumors [117].Phensuximide: This anticonvulsant has not been previously investigated as an antitumor agent. However, network pharmacology and drug repurposing studies have identified it as a potential therapeutic agent for adrenal cortical [118] and breast [119] carcinomas, including a highly invasive form of the latter [120].

In the verification block of the study, we confirmed that SP600125 effectively suppressed GMT in GBM cells induced by TGF-β (Figure 11) and chemical hypoxia (Figure 12). Specifically, SP600125 reversed the morphological changes (Figure 11A and Figure 12A and Appendix A), vasculogenic mimicry formation (Figure 11C and Figure 12B), and mesenchymal gene activation (Figure 11D and Figure 12C) in both models. Moreover, SP600125 effectively suppressed the FBS-stimulated transwell migration of TGF-β-induced GBM cells (Figure 11B).

The association between GMT and M1-like macrophage signatures, initially identified through in silico analysis (Figure 4C), was partially validated in this study. We demonstrated that the induction of GMT by either TGF-β1 or chemical hypoxia upregulated the expression of pro-inflammatory cytokines IL-6 (Figure 11G and Figure 12D) and IL-1β (Figure 12D), and these changes were susceptible to SP600125 (Figure 11G and Figure 12D,E). These findings indicate a close relationship between GMT and GBM cells’ ability to shape a distinct immune microenvironment, thereby promoting malignant progression. This interpretation is consistent with published studies [121,122,123].

SP600125 is a well-known JNK1/2 inhibitor characterized by low kinase selectivity and the ability to interact with other protein targets [124]. Based on connectivity map analysis (Figure 9C), we conclude that the pronounced anti-GMT activity of SP600125 is not primarily due to its direct action on JNK1/2, as most other JNK inhibitors exerted the opposite effect on the GMT signature. Instead, its activity is likely explained by multi-target properties, specifically interactions with JNK1, MELK, and FLT3. Indeed, Liu et al. previously demonstrated a key regulatory role of MELK in ankyrin repeat domain protein 22 (ANKRD22)-mediated GMT in GBM cells [125]. FLT3 also plays an important role in the progression of intracranial tumors: mutations in FLT3 correlate with worse survival in glioma patients [126], while pharmacological inhibition of FLT3 reduces meningioma cell motility [127]. A particularly intriguing finding of this study is that JNK1, but not JNK2, is negatively associated with the GMT signature (Figure 9C). This observation presents an interesting therapeutic opportunity. However, due to the high homology of the ATP-binding sites of JNK1 and JNK2 (98% similarity) [124], developing a selective JNK1-targeting anti-GMT compound poses a significant challenge.

Despite its poor kinase selectivity and off-target effects, intraperitoneal administration of SP600125 demonstrated significant antitumor efficacy in vivo in a murine model of Lewis lung carcinoma and DU145 prostate carcinoma xenografts [128]. Additionally, SP600125 markedly potentiated the antitumor effect of semisynthetic anhydrophytosphingosine C-2 in EJ bladder carcinoma xenografts [129], indicating SP600125’s ability to accumulate in tumor tissue. Furthermore, SP600125 was effective against Parkinson’s disease and encephalopathy induced in mice by 1-methyl-4-phenyl-1,2,3,6-tetrahydropyridine [130] and cecal ligation and puncture [131], respectively. These findings confirm that SP600125 can cross the blood–brain barrier and is well-tolerated in brain tissue.

The absence of significant systemic toxicity in mouse models [128,129,130,131] suggests that despite its limitations, SP600125 could serve as a platform for developing novel antitumor agents. Indeed, its derivative CC-401 has been evaluated in clinical trials for acute myeloid leukemia (NCT00126893) [132]. Another derivative, CC-930, which shows a connectivity profile similar to SP600125 (Figure 9C), was discontinued in Phase 2 due to an increased risk of hepatotoxicity (NCT01203943) [133]. Clearly, further development of SP600125 as a structural platform for new anti-GMT agents requires molecular derivatization to mitigate its adverse effects. Promising strategies include introducing alkyl substituents at the N2 position [134,135], cleavage of the cyclohexanone ring followed by introduction of substituents [136], or conjugation of the molecule with targeting moieties [137].

We acknowledge that this study has several limitations. Although transcriptomic analysis suggests a promising regulatory role for the revealed hub gene signature in GMT induction (Figure 5A), experimental verification of this issue is necessary. Our lead compound selection was based on its structural properties (Figure 7B,D and Figure 8A) and interactions with the GMT-related hub gene signature (Figure 9A,B), but successful repurposing requires consideration of additional features such as patent restrictions, clinical approval status, and treatment cost. Furthermore, although JNK signaling is tightly linked to cancer progression, SP600125 indiscriminately inhibits phosphorylation of all JNK substrates, which determines its low tumor specificity [138]. In view of this, it is crucial to elucidate the molecular mechanisms of SP600125’s anti-GMT activity and to identify its key protein targets. Future research should focus on the derivatization of SP600125 to improve its selectivity and experimental testing of the remaining hits (vemurafenib, FG-7142, dibenzoylmethane, and phensuximide) for GMT inhibitory properties. Finally, the application of state-of-the-art machine learning approaches, which enable the analysis of global transcriptomic data beyond the limited gene sets typically used in Connectivity Map analysis [106,107] could significantly complement the results presented here.

## 4. Materials and Methods

### 4.1. Bioinformatics

#### 4.1.1. Data Acquisition

The Cancer Genome Atlas—Glioblastoma multiforme (TCGA-GBM) dataset, which includes transcriptomic profiles and survival information for 168 patients, was sourced from the Genomic Data Commons (GDC) data portal (https://portal.gdc.cancer.gov/, accessed on 26 December 2023). The datasets “mRNAseq_693” and “mRNAseq_325” were obtained from the Chinese Glioma Genome Atlas (CCGA) (https://www.cgga.org.cn/, accessed on 10 September 2024) and subsequently merged. To address batch effects, the “Combat_seq” function from the sva package (v3.54.0) [139] in R was used. Subsequently, only samples with confirmed GBM histology and corresponding survival information were retained, resulting in a total of 218 patients. The Gravendeel dataset was accessed via the GlioVis portal (http://gliovis.bioinfo.cnio.es/, accessed on 10 September 2024). After filtering for GBM patients with survival data, 155 samples remained. The RNA-Seq data from TCGA-GBM and CGGA were normalized using the variance stabilizing transformation (VST) method, while the microarray data from the Gravendeel dataset underwent log normalization. The GSE192710 dataset was retrieved from the Gene Expression Omnibus database (https://www.ncbi.nlm.nih.gov/geo/, accessed on 11 January 2024) as raw read counts for subsequent differential expression analysis.

#### 4.1.2. Gene Set Enrichment Analysis

The gene signatures for mesenchymal GBM (M2122), proneural GBM (M2115), EMT (M5930), hypoxia (M5891), TGF-β signaling (M5896), and inflammatory response (M5932) were retrieved from the Molecular Signatures Database (MSigDB). The enrichment of signatures was evaluated in TCGA-GBM patient tumors using Gene Set Variation Analysis (GSVA) from the GSVA package (v2.0.7) in R [140]. The GMT score was calculated by subtracting the proneural score from the mesenchymal score, as previously described [33]. The samples were divided according to the median GMT score into two subgroups, designated as “GMT high” and “GMT low”. Subsequently, Gene Set Enrichment Analysis (GSEA) was conducted using GSEA software (v4.3.3) [141] to assess the enrichment of gene signatures of EMT, hypoxia, TGF-β signaling, and inflammatory response in the “GMT high” and “GMT low” subgroups. The significance criteria were set at a nominal *p*-value < 0.05 and FDR < 25%.

#### 4.1.3. Text Mining Analysis

The co-occurrence of hub genes with the terms “epithelial–mesenchymal transition OR EMT”, “glioblastoma”, “metastasis”, and “invasion” was analyzed in the MEDLINE database using the default settings of GenCLiP3 (http://cismu.net/genclip3/analysis.php, accessed on 3 November 2024) [142]. Terms were considered related if the number of articles found exceeded three. A similar analysis for potential GMT inhibitors was performed using the LitSense search system (https://www.ncbi.nlm.nih.gov/research/litsense/, accessed on 3 November 2024), which conducts co-word analysis on sentences from the PubMed and PubMed Central databases [143] A manual review of sentences was carried out to exclude potential misinterpretations of the terms “EMT” (e.g., “emergency medical technician”, “endometriosis”) and “invasion” (e.g., “invasive procedure”, “invasive infections”). Terms were considered related if the number of sentences exceeded five.

#### 4.1.4. Survival Analysis

A survival analysis was conducted using the survival (v3.8.3) and survminer (v0.5.1) packages [144,145] of R. Genes associated with prognosis were selected using univariate Cox proportional hazards regression analysis, with a *p*-value < 0.05 considered significant. The patient cohorts were stratified by median GMT score or median expression of prognosis-related genes. Subsequently, Kaplan–Meier curves were plotted, and differences were evaluated using the log-rank test, with a significance threshold of *p* < 0.05.

#### 4.1.5. Immune Cell Quantification

An immune cell type enrichment analysis was performed on TCGA-GBM samples using the xCell package (v1.1.0) [146] in R. Subsequently, a literature review was conducted to identify immune subtypes that had been experimentally validated for GBM.

#### 4.1.6. Weighted Gene Co-Expression Network Analysis (WGCNA)

WGCNA was conducted using the WGCNA package (v1.73) [38] in R. Outlier genes were filtered using the “goodSamplesGenes” function. A soft-threshold power β of 12 was chosen using the “pickSoftThreshold” function to achieve a scale-free topology. A co-expression network was created using the “blockwiseNodules” function, after which similar modules were merged by cutting the dendrogram at a height of 0.25. Finally, Pearson correlation analysis was performed to examine the relationship between the module eigengenes and the GMT score or EMT enrichment.

#### 4.1.7. Differential Expression Analysis

The transcriptomes of E2 and G7 cells, which represent proneural and mesenchymal GBM, respectively, were obtained from GSE192710. Genes with increased expression in G7 cells compared to E2 cells were identified using differential expression analysis in the DESeq2 R package (v1.46.0) [147]. The analysis employed a Wald test with significance thresholds of a Bonferroni-corrected *p*-value < 0.05 and a log fold change (logFC) ≥ 1.

#### 4.1.8. Gene Network Analysis

Upregulated differential expressed genes (DEGs) were used to construct a gene association network in Cytoscape (v3.10.3) with the STRING plugin (v2.2.0) [148], applying a confidence score > 0.7 and no additional interactions as cutoff criteria. Nodes in the network were ranked by topological features using the cytoHubba plugin (v0.1) [149], with centrality scores calculated by averaging the ranks of Degree, Maximal Clique Centrality (MCC), Maximum Neighborhood Component (MNC), Edge Percolation Component (EPC), Closeness, and Radiality. Subsequently, 30 hub genes were identified based on their centrality scores and inclusion in the GMT-associated co-expression module as analyzed by WGCNA.

GMT inhibitors, their protein targets, and interactions were obtained from the STITCH database v5.0 (confidence score > 0.7; interaction source: text mining, experiments, and databases) [150]. They were then mapped to a hub gene network using the STRING plugin in Cytoscape. The most interconnected GMT inhibitors in the network were identified via cytoHubba, using the same procedure as for hub genes.

#### 4.1.9. Connectivity Map Analysis

The CLUE platform (https://clue.io/, accessed on 11 January 2024) was used to compare the hub gene signature with expression profiles of nine tumor cell lines treated by small molecules and gene knockdown [22]. According to CLUE Connectopedia, connectivity scores ≥90 or ≤−90 indicate strong similarity or dissimilarity, respectively. We selected small molecules with scores ≤−90 as potential GMT inhibitors. Their mechanisms of action were investigated using CLUE data and a literature review. To assess the similarity between potential GMT inhibitors and FDA-approved or investigational GBM drugs, connectivity scores were analyzed using the Touchstone dataset. For SP600125, its mechanism was analyzed by comparing the expression profiles of JNK inhibitors and SP600125 target genes with the GMT signature.

#### 4.1.10. Chemoinformatics

The blood-barrier permeability of potential GMT inhibitors was predicted using AlzPlatform (https://www.cbligand.org/AD/, accessed on 15 January 2024) [151] and PreADMET (https://preadmet.qsarhub.com/, accessed on 15 January 2024). The top 30 hit compounds from both platforms were intersected and assessed for their specificity as substrates of P-glycoprotein (P-gp) using the Vienna LiverTox Workspace (https://livertox.univie.ac.at/, accessed on 15 January 2024) [152], SwissADME (https://www.swissadme.ch/, accessed on 15 January 2024) [153], and ADMETlab (https://admet.scbdd.com/, accessed on 15 January 2024) [154]. Doxorubicin and temozolomide served as negative and positive controls, respectively, due to their contrasting BBB permeability and P-gp specificity. Further details on these analyses can be found in an earlier publication [155]. The structural similarity of potential GMT inhibitors with FDA-approved and investigational GBM drugs was evaluated using the Tanimoto score in ChemBioServer 2.0 (https://chembioserver.vi-seem.eu/, accessed on 12 October 2024) [156].

### 4.2. Chemicals and Reagents

CoCl_2_ (232696-5G) was obtained from Sigma-Aldrich (St. Louis, MO, USA). Fresh stock solutions of CoCl_2_ (25 mM) in deionized water were prepared each time before addition to cells. Human recombinant TGF-β1 (CYT-716) was purchased from ProSpec-Tany TechnoGene Ltd. (Ness-Ziona, Israel). SP600125 (ab120065) was obtained from Abcam (Waltham, MA, USA), dissolved in DMSO at a concentration of 10 mM, and stored at −20 °C before use. M-MuLV–RH First Strand cDNA Synthesis Kit and BioMaster SYBR Blue reagent kit were obtained from Biolabmix (Novosibirsk, Russia). Primary rabbit anti-human N-cadherin (ab76011) and vimentin (ab92547) antibodies, along with secondary HRP-conjugated (ab205718) and Alexa Fluor 488-conjugated (ab150077) anti-rabbit IgG antibodies, were purchased from Abcam (Waltham, MA, USA).

### 4.3. Cell Lines

Human GBM U87 and U118 and mouse GBM EPNT-5 cell lines were obtained from the Russian Culture Collection (Institute of Cytology of the Russian Academy of Sciences (RAS), St. Petersburg, Russia). Mouse GBM GL261 cells and human non-transformed hFF3 foreskin fibroblasts were kindly provided by Dr. Alexey Stepanenko (Department of Neurobiology, V. P. Serbsky National Medical Research Center for Psychiatry and Narcology, Moscow, Russia) and Dr. Olga A. Koval (Institute of Chemical Biology and Fundamental Medicine of the Siberian Branch of RAS, Novosibirsk, Russia), respectively. All cells were maintained in DMEM medium supplemented with 10% fetal bovine serum (FBS; Dia-M, Moscow, Russia) and 1% antibiotic-antimycotic solution (100 U/mL penicillin, 100 μg/mL streptomycin, 0.25 μg/mL amphotericin B (Central Drug House Pvt. Ltd., New Delhi, India)) at 37 °C and 5% CO_2_ in a humidified atmosphere. During experimental procedures, the medium was replaced with DMEM without additives to avoid effects associated with growth factors, antibiotics, and antimycotics.

### 4.4. Biological Evaluations

#### 4.4.1. Cell Viability Analysis

U87, U118, GL261, EPNT-5, and hFF3 cells were seeded at 10,000 cells/well in 96-well plates and allowed to adhere overnight. SP600125 (0–100 μM) was then added to the cells for 24 h and 48 h, after which cell viability was assessed using the MTT assay as described previously [157].

#### 4.4.2. Wound Healing Assay

U87, U118, and GL261 cells were placed in 24-well plates at densities of 250,000, 300,000, and 150,000 cells per well, respectively. When the cells reached approximately 80% confluence, two perpendicular scratches were made across the cell layer using a 100 μL pipette tip. Detached cells were then removed by washing with PBS. Subsequently, the cells were co-incubated with low concentrations (0–2.5 μM) of SP600125 for 48 h. Alternatively, the cells were pre-incubated with a high concentration (20 μM) of SP600125 for 30 min and maintained in serum-free medium for the remainder of the experiment. Scratches were photographed at 0, 24, and 48 h using an EVOS XL Core microscope (Thermo Fisher Scientific, Waltham, MA, USA). Wound closure was calculated by assessing the wound area at each time point using ImageJ (v2.14.0) and normalizing it to 0 h.

#### 4.4.3. Assessment of Cell Morphology

U87 and U118 cells were seeded at 5000 cells/well in 96-well plates and allowed to adhere overnight. The cells were pretreated with SP600125 (20 μM) for 30 min prior to incubation with TGF-β1 (50 ng/mL) or CoCl_2_ (400 μM) for 48 h. The photographs were obtained using an EVOS XL Core microscope. Morphology was analyzed for 200 cells per group using the parameters of the area, circularity (circ), aspect ratio (AR), roundness (round), and solidity in ImageJ. In the main text, we focused on area and AR, while the other parameters, along with between-group statistical comparisons, are provided in Appendix A.

#### 4.4.4. Transwell Migration Assay

500,000 U87 cells were allowed to attach to a 6-well plate overnight. Following a 30 min pre-treatment with SP600125 (20 μM), the cells were incubated with TGF-β1 (50 ng/mL) for 48 h. Thereafter, they were detached using TrypLE Express (Gibco, Grand Island, NY, USA) and placed at 50,000 cells in the upper chamber of a transwell plate in quadruplicate. The migration of cells towards 10% FBS placed in the lower chamber was analyzed using the xCELLigence RTCA DP system (ACEA Biosciences Inc., San Diego, CA, USA), as previously described [14]. In the main text, we focused on the 24 h endpoint, while Appendix A provides ΔNCI and between-group statistical comparison for each hour of the experiment.

#### 4.4.5. Vasculogenic Mimicry Analysis

300,000 U87 cells were placed in a 24-well plate. The next day, cells were pretreated with SP600125 (20 μM) for 30 min and then supplemented with TGF-β1 (50 ng/mL) or CoCl_2_ (400 μM). Photographs were taken by the EVOS XL Core microscope at 48 h of incubation. Tubular structures were analyzed by estimating the mean mesh size using the Angiogenesis Analyzer plugin for ImageJ [158].

#### 4.4.6. Western Blot

U87 cells (100,000 cells/well) were pretreated with SP600125 (20 μM) for 30 min and incubated with TGF-β1 (50 ng/mL) for 48 h. Cell lysate was prepared using Laemmli buffer (Sigma-Aldrich, St. Louis, MO, USA). Further experimental procedures were carried out as described previously [157].

#### 4.4.7. RT-qPCR

Following 30 min treatment with SP600125 (20 μM) and 48 h incubation with either TGF-β1 (50 ng/mL) or CoCl_2_ (400 μM), U87 cells were lysed with TRIzol Reagent (Ambion, Austin, TX, USA). The cDNA synthesis and RT-qPCR were conducted using the M-MuLV–RH First Strand cDNA Synthesis Kit and BioMaster SYBR Blue reagent kit, respectively, as described previously [14]. Appendix A contains primer sequences for mesenchymal markers and the housekeeping gene *HPRT*.

#### 4.4.8. Immunofluorescence

150,000 U87 cells were seeded onto glass coverslips in 24-well plates. The next day, cells were treated with SP600125 (20 μM) for 30 min, followed by TGF-β1 (50 ng/mL) for 48 h. Cells were fixed with 4% formaldehyde, then incubated with anti-vimentin antibody (1:250) in permeabilization buffer (PBS with 0.1% Triton X-100 and 5 mg/mL BSA) for 1 h. AlexaFluor 488-conjugated secondary antibody (1:500) was applied in staining buffer (5 mg/mL BSA in PBS) for 1 h in the dark. Nuclei were stained by DAPI (1 μg/mL in PBS). Each staining step was followed by two washes. Coverslips were mounted on slides with Fluoromount-G (Thermo Fisher Scientific, Rockford, IL, USA). Images were captured using a LSM710 confocal microscope (Zeiss, Oberkochen, Germany) at ×200 magnification.

#### 4.4.9. Enzyme-Linked Immunosorbent Assay (ELISA)

U87 cells were seeded in 96-well plates at 10,000 cells/well and allowed to adhere overnight. They were treated with SP600126 (20 μM) for 30 min, followed by CoCl_2_ (400 μM) for 48 h, after which the culture medium was collected. ELISA was performed using the Interleukin-6-EIA-BEST kit (Vector-Best, Novosibirsk, Russia). Four-fold diluted medium was incubated for 2 h in plates coated with IL-6 monoclonal antibodies, then with biotinylated IL-6 antibodies for 1 h, followed by streptavidin-HRP for 30 min. Each step was performed at 500 rpm and 37 °C on a thermostatic shaker (ELMI ST-3L, Riga, Latvia), followed by five washes with Tween-PBS. The colorimetric reaction was developed by incubating samples with tetramethylbenzidine for 25 min in the dark and stopped by adding 0.5 M sulfuric acid. Absorbance was measured at 450 nm using a Multiscan RC plate reader (Thermo LabSystems, Helsinki, Finland). IL-6 concentration was determined using a calibration curve from standards of known concentration.

### 4.5. Data Analysis and Visualization

Data analysis and visualization was conducted using R (v4.4.2) and RStudio (v.2024.12.1). A pairwise group comparison was conducted using the Student’s *t*-test and Wilcoxon rank sum test from the rstatix (v0.7.2) package in R. Spearman correlation analysis was performed, and the results were visualized using the Hmisc (v5.2.3) and corrplot (v0.95) packages in R, respectively. Additional visualization tools included ggplot2 (v4.0.0) [159], ComplexHeatmap (v2.22.0) [160], ggvenn (v0.1.10) [161], networkD3 (v0.4.1) [162], and plotly (v4.11.0) [163].

## 5. Conclusions

This complex study yielded three sets of results that are novel and of interest for molecular oncology and drug development. First, a novel gene signature susceptible to GMT induction in GBM cells, occupying hub positions in the GMT-related regulome, was identified. This list included 31 genes, a significant proportion of which were found to be associated with the inflammatory response, suggesting an association of GMT with the deployment of cytoprotective immune-related programs in GBM cells. Second, using the GMT-related hub gene signature, a number of drug candidates with anti-GMT potency were predicted, including SP600125, vemurafenib, FG-7142, dibenzoylmethane, and phensuximide. Given their negative effect on GMT-related hub gene expression and published data confirming their ability to suppress EMT in non-brain tumor cells, the effect of these compounds on GMT in GMB cells requires further detailed studies. Third, for the first time, SP600125 was shown to effectively suppress GMT induced by TGF-β and chemical hypoxia in GBM cells, inhibiting morphological changes, motility, vasculogenic mimicry, and the expression of mesenchymal markers. These results not only expand the understanding of the anti-tumor potential of SP600125 but also confirm the applicability of the connectivity map-based drug repurposing approach in the search for novel anti-GBM drug candidates.

## Figures and Tables

**Figure 1 ijms-26-09772-f001:**
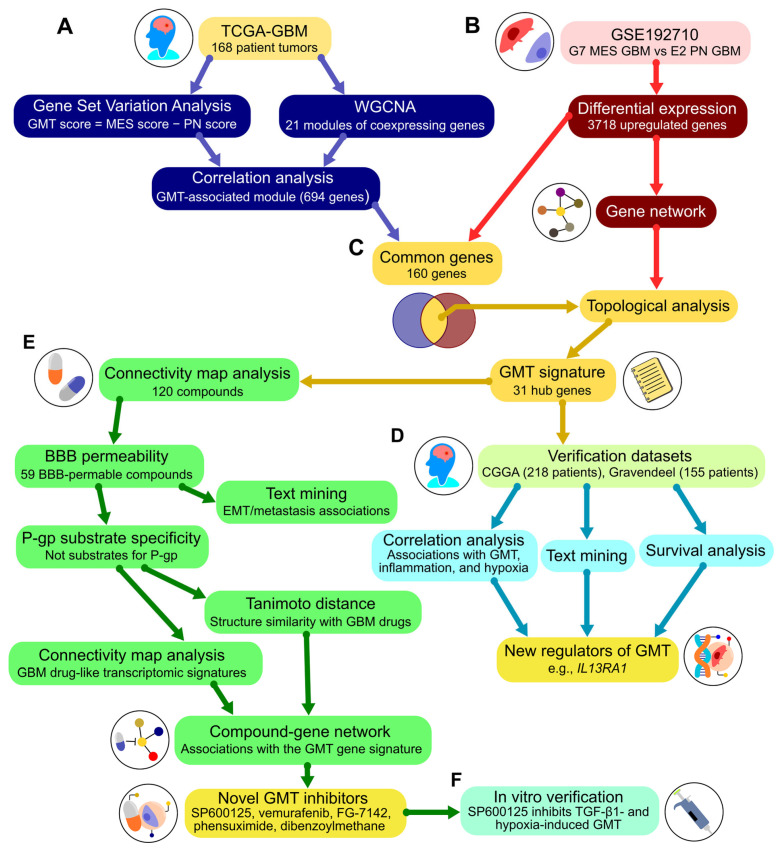
Workflow overview. (**A**) Analysis of The Cancer Genome Atlas (TCGA)-GBM dataset. The enrichment of mesenchymal and proneural gene signatures was assessed using gene set variation analysis to calculate the GMT score. Modules of co-expressed genes were identified via weighted correlation network analysis. The module correlating with the GMT score was carried forward to step (**C**). (**B**) Analysis of the GSE192710 dataset. Differential expression analysis compared mesenchymal subtype G7 cells to proneural subtype E2 cells. Upregulated genes were used to construct a gene network with STRING and included in the subsequent analysis in (**C**). (**C**) Formation of the GMT signature. The positions of genes common to both the GMT-associated module (**A**) and the upregulated genes (**B**) were analyzed in the gene network (**B**) using cytoHubba. This analysis identified 31 hub genes included in the GMT signature. (**D**) Verification of the GMT signature. Additional clinical datasets (Chinese Glioma Genome Atlas (CGGA) and Gravendeel) were analyzed for associations of hub genes with the GMT signature, the literature data, and survival outcomes. This analysis identified new regulators of GMT. (**E**) Drug repurposing. The Connectivity Map platform was used to find small molecules with negative connectivity to the GMT signature. Further screening assessed blood-brain permeability and P-glycoprotein substrate specificity using chemoinformatic tools. Text mining evaluated compound associations with GMT-related terms in the literature. Tanimoto distance and Connectivity Map analyses examined structural similarity and biological activity compared to known GBM drugs. The screening identified hit compounds, which were integrated into a network with their primary targets and the GMT signature using the Search Tool for Interactions of Chemicals (STITCH). Topological analysis of this drug-gene network revealed novel potential GMT inhibitors. (**F**) In vitro verification of the GMT inhibitory effect of the hit compound was conducted using cell models of GMT induced by TGF-β1 or hypoxia. MES, mesenchymal; PN, proneural; WGCNA, weighted correlation network analysis; BBB, blood–brain barrier; P-gp, p-glycoprotein.

**Figure 2 ijms-26-09772-f002:**
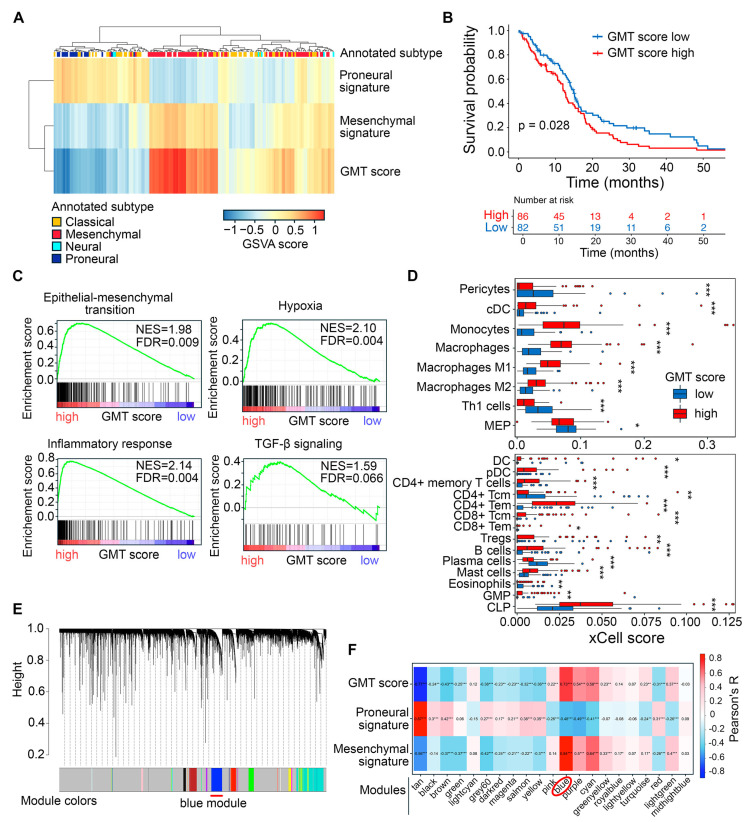
Identification of a GMT-associated gene cluster in TCGA-GBM tumor transcriptomes. (**A**) Characterization of GMT status in patient tumors. GSVA was used to evaluate the enrichment of Verhaak’s proneural and mesenchymal gene signatures in each tumor sample, with the GMT score calculated as the difference between these enrichments. The color legend at the top of the heatmap indicates the tumor subtype (classical, neural, proneural, or mesenchymal) extracted from the TCGA-GBM cohort metadata. (**B**) Kaplan–Meier survival curves for patients stratified by median GMT score. (**C**) GSEA of gene signatures for EMT, hypoxia, inflammatory response, and TGF-β signaling in tumors with low and high GMT scores. (**D**) Enrichment of immune cell types in GMT score-stratified tumors assessed using xCell. Statistical significance was tested by the Wilcoxon test: *, **, *** indicate *p*-values < 0.05, <0.01, and <0.001, respectively. (**E**) Clustering dendrogram of TCGA-GBM tumors generated by WGCNA, with colors representing modules of co-expressed genes. Red line under the dendrogram indicating the blue module, identified as tightly connected with GMT in subsequent analysis (**F**) Module–trait relationship identified by Pearson’s correlation analysis between module eigengenes (weighted average expression of each gene set) and GMT score, as well as proneural and mesenchymal gene set enrichments. The blue module, showing the strongest correlation with GMT score and mesenchymal signature enrichment, is highlighted by a red circle below the heatmap. *, **, *** indicate *p*-values < 0.05, <0.01, and <0.001, respectively.

**Figure 3 ijms-26-09772-f003:**
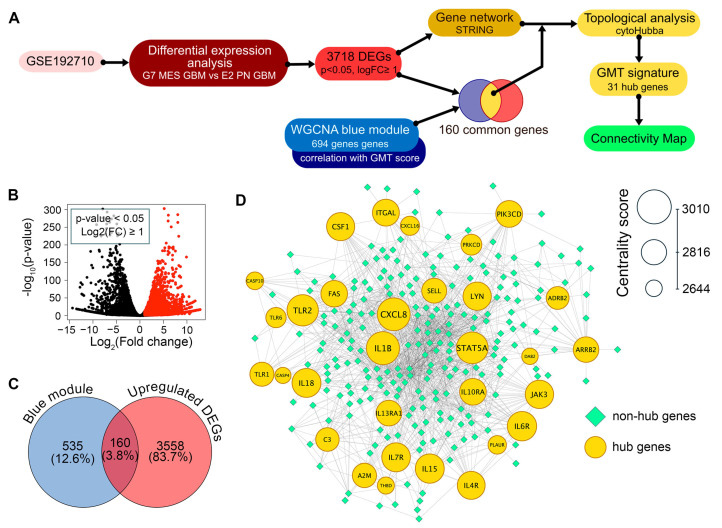
Identification of the GMT gene signature. (**A**) Workflow. Differential expression analysis was performed on the GSE192710 dataset to find genes upregulated in G7 mesenchymal GBM cells compared to E2 proneural GBM cells (*p*-value < 0.05, log2(fold change) ≥ 1). These genes were used to construct a gene network using STRING. Upregulated genes overlapping with the WGCNA blue module (Figure 2E,F) were selected, and their positions within the network were analyzed using cytoHubba. This analysis identified 31 hub genes that form the GMT signature, which were then used in Connectivity Map analysis. (**B**) A volcano plot displaying the results of differential expression analysis between mesenchymal G7 cells and proneural E2 cells from the GSE192710 dataset. Genes that are significantly upregulated (*p*-value < 0.05 and log_2_(fold change) ≥ 1) are represented as red dots. (**C**) Intersection of the upregulated DEGs from GSE192710 and the GMT-associated blue module of WGCNA. (**D**) The GMT regulatory network, in which the top-30 hub genes are color-coded and sized by centrality score.

**Figure 4 ijms-26-09772-f004:**
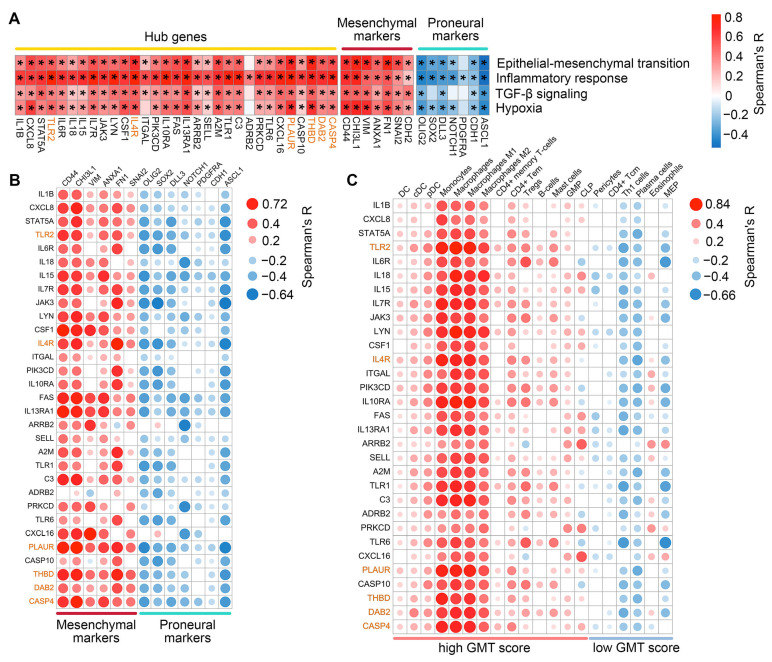
Associations of the hub genes with hypoxia, inflammation, and expression of proneural and mesenchymal markers. (**A**) Spearman’s correlation between hub gene expression and GSVA enrichment scores for EMT, inflammatory response, TGF-β signaling, and hypoxia in TCGA-GBM, with significant results marked by an asterisk (* *p* < 0.05). (**B**) Spearman’s correlation between hub gene expression and established markers of mesenchymal and proneural GBM subtypes in TCGA-GBM. Only significant results (*p* < 0.05) are shown. (**C**) Correlation between hub gene expression and immune cell enrichment scores in TCGA-GBM. Only significant results (*p* < 0.05) are shown. Genes belonging to Verhaak’s mesenchymal signature are highlighted in orange.

**Figure 5 ijms-26-09772-f005:**
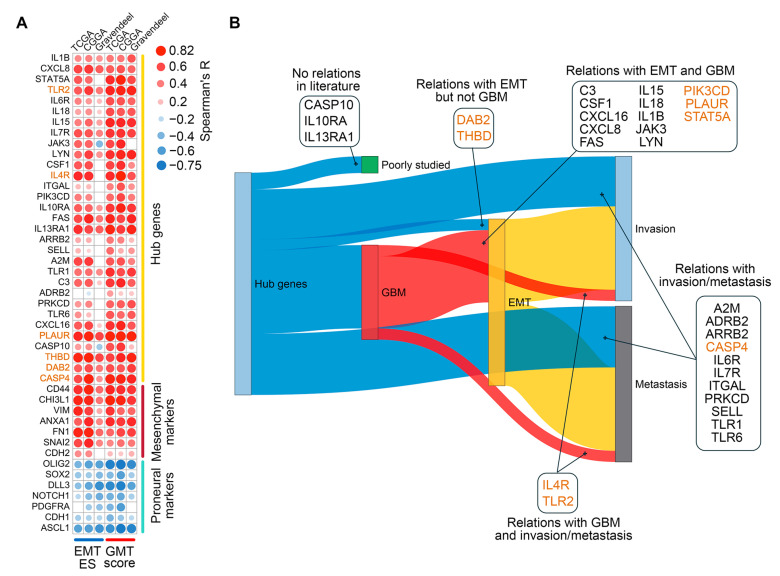
Validation of the regulatory role of hub genes in GMT. (**A**) Spearman’s correlation analysis of hub gene expression with GMT scores and EMT enrichment scores (EMT ES) across the TCGA-GBM, CGGA, and Gravendeel datasets. Only significant results (*p* < 0.05) are shown. (**B**) Relationships of hub genes with GBM (red), EMT (yellow), invasion (soft blue), and metastasis (gray) were identified in the literature using GenCLiP3. Genes belonging to Verhaak’s mesenchymal signature are highlighted in orange.

**Figure 6 ijms-26-09772-f006:**
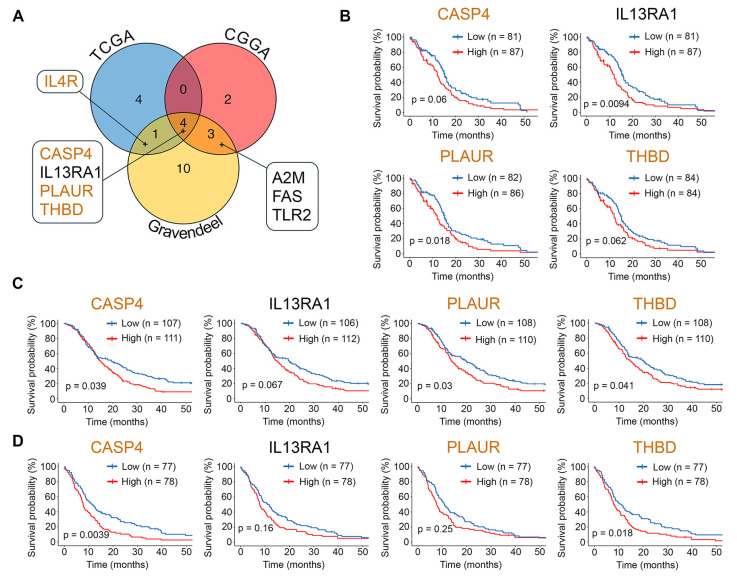
Relationship between hub gene expression and survival outcomes in GBM. (**A**) Intersection of prognosis-related hub-genes identified by univariate Cox analysis (HR > 1) in the TCGA-GBM, CGGA, and Gravendeel datasets. (**B**–**D**) Kaplan–Meier survival curves for patients from the TCGA-GBM (**B**), CGGA (**C**), and Gravendeel (**D**) datasets, stratified by median expression levels of prognosis-related hub genes. *P*-values were determined by the log-rank test. Genes belonging to Verhaak’s mesenchymal signature are highlighted in orange.

**Figure 7 ijms-26-09772-f007:**
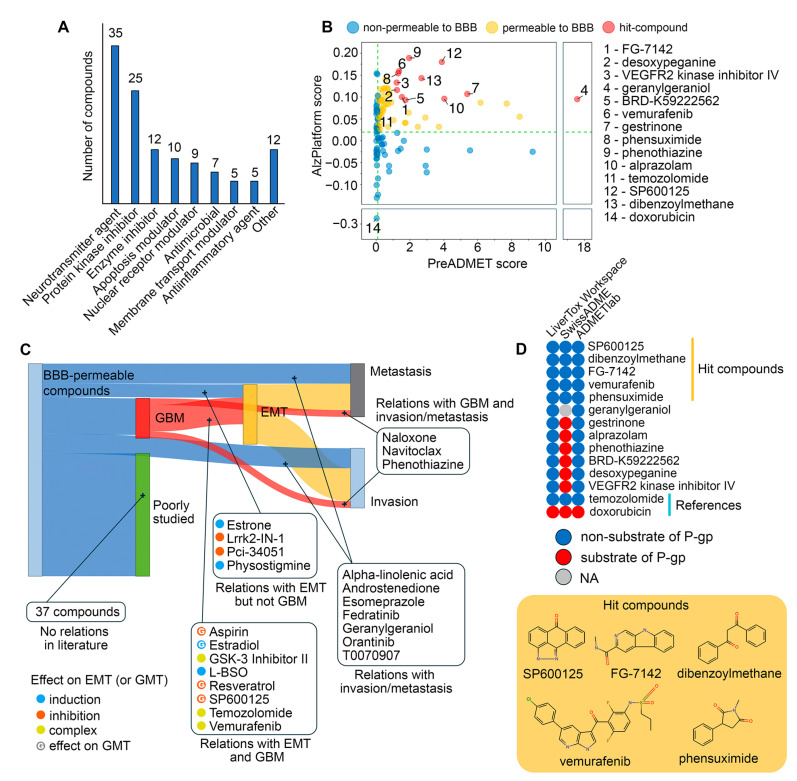
In silico repurposing of small molecule drugs for GMT targeting. (**A**) Mechanisms of action of the repurposed drugs identified through connectivity map analysis on the CLUE platform, based on opposing their expression signatures to GMT-associated hub genes. (**B**) Predictions of BBB-permeability for the repurposed drugs using AlzPlatform and PreADMET, with temozolomide and doxorubicin serving as negative and positive controls, respectively. (**C**) Relationships between BBB-permeable drugs and GBM (red), EMT (yellow), invasion (soft blue), and metastasis (gray) identified in the literature through LitSense. (**D**) Predictions of P-gp substrate specificity for BBB-permeable drugs using LiverTox Workspace, SwissADME, and ADMETlab, with doxorubicin and temozolomide serving as positive and negative controls, respectively. Structures of the hit compounds are shown below.

**Figure 8 ijms-26-09772-f008:**
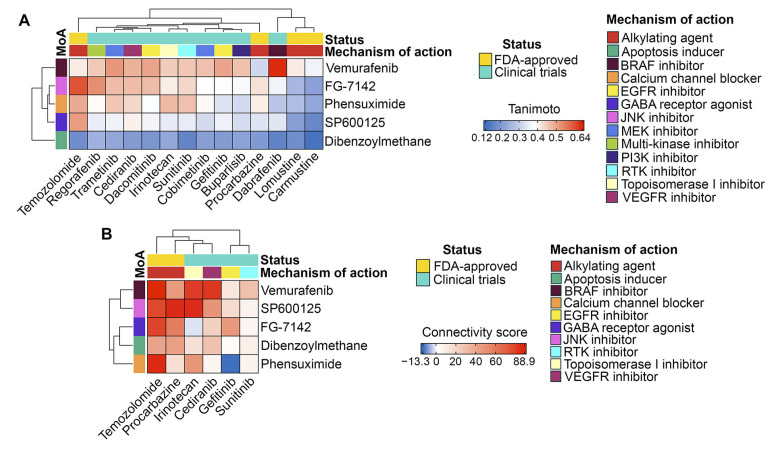
Comparison of the hit compounds with approved or clinically investigated GBM drugs. (**A**) Tanimoto similarity matrix of the hit compounds with FDA-approved and investigational GBM drugs. (**B**) Associations between the expression signatures of the hit compounds and GBM drugs identified by the CLUE platform.

**Figure 9 ijms-26-09772-f009:**
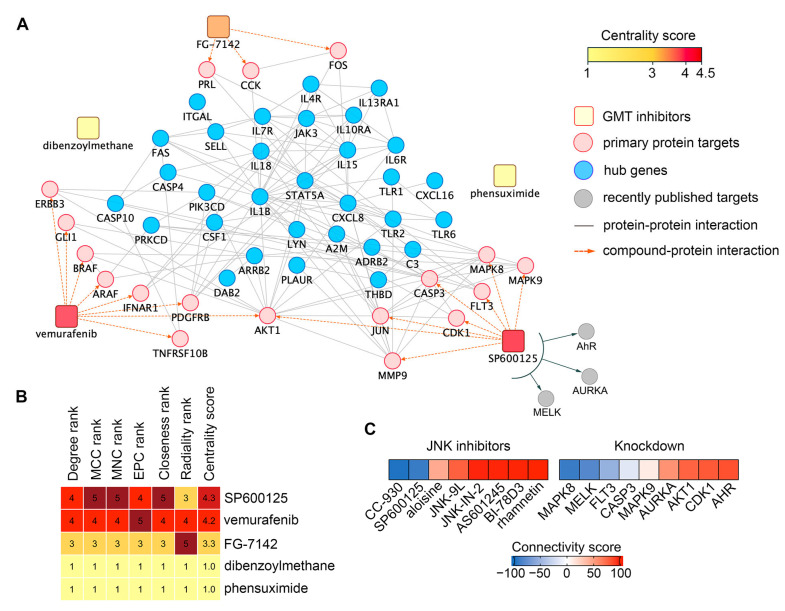
In silico analysis of the anti-GMT activity of SP600125. (**A**) Compound-protein network illustrating connections between hit compounds (rounded rectangles, color-coded by centrality score), their primary protein targets (pink circles), and hub genes (blue circles). (**B**) Assessment of compound importance in the network using six topological algorithms. (**C**) Similarity between the GMT-associated gene signature and gene expression profiles from tumor cells treated with JNK inhibitors or with knockdown of genes encoding SP600125’s protein targets. Analysis was performed using the CLUE platform.

**Figure 10 ijms-26-09772-f010:**
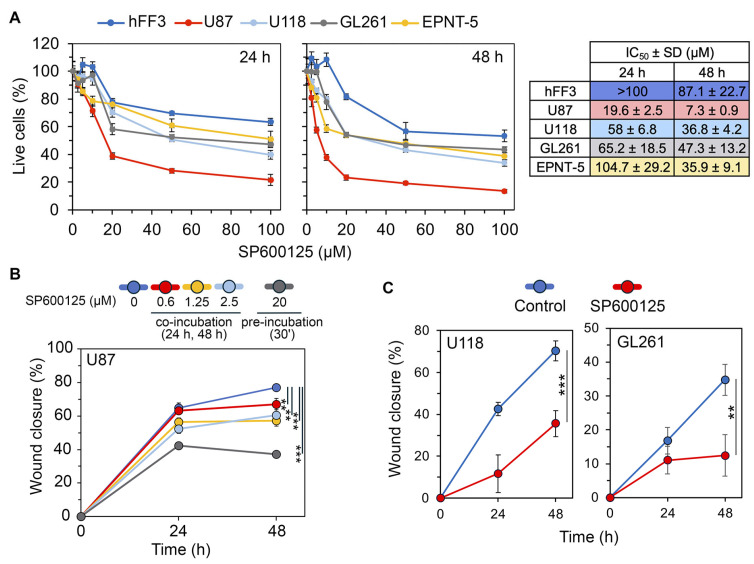
Selection of an effective incubation scheme with SP600125. (**A**) Viability of human GBM cells (U87 and U118), mouse GBM cells (EPNT-5 and GL261), and human non-malignant cells (hFF3) after 24 h and 48 h of incubation with SP600125 assessed by MTT assay. IC50 values are shown on the right. (**B**) Impact of various SP600125 incubation regimens on U87 cell motility in a wound healing assay. U87 cells were subjected to low doses (0–2.5 μM) for 48 h or were pretreated with high doses (20 μM) for 30 min and then left untreated for the rest of the time. (**C**) Effect of SP600125 preincubation on the motility of U118 and GL261 cells. Representative wound images at each time point are provided in Appendix A. Data are represented as mean ± standard deviation (SD). Statistical significance was determined by comparison with control groups using a *t*-test. **, *** indicate *p*-values <0.01 and <0.001, respectively.

**Figure 11 ijms-26-09772-f011:**
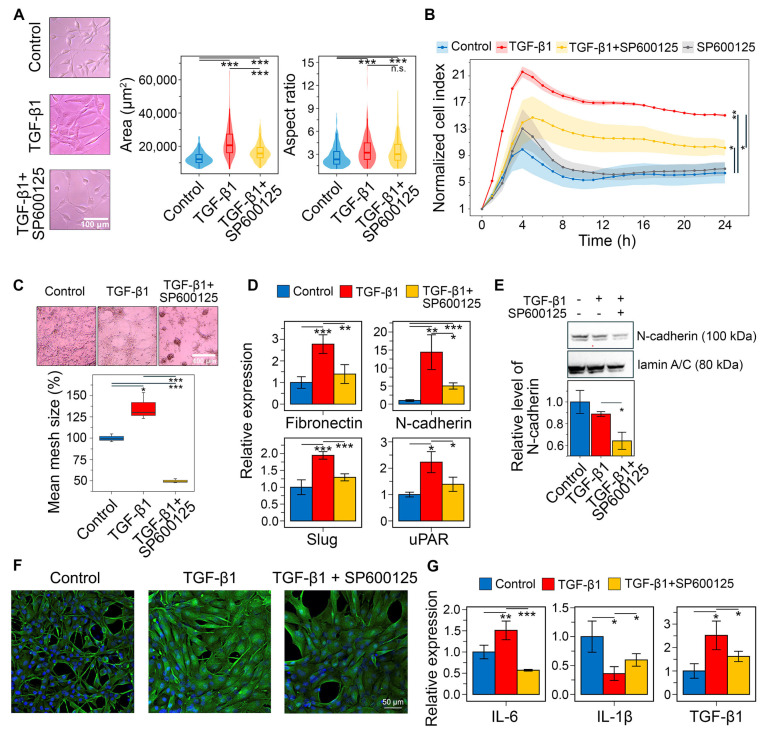
Influence of SP600125 on TGF-β1-induced EMT in U87 GBM cells in vitro. (**A**) U87 cell morphology after 30 min of SP600125 (20 μM) induction and 48 h of TGF-β1 (50 ng/mL) stimulation. Marginal graphs in the upper and right panels show the distribution of cells by area and aspect ratio, respectively, with dashed lines indicating mean values. The image on the left was taken at ×200 magnification. (**B**) Transwell migration of U87 cells after SP600125 pretreatment and TGF-β1 activation. (**C**) The effect of SP600125 on vasculogenic mimicry of TGF-β1-treated U87 cells. The microscopic image was taken at ×40 magnification. (**D**) Relative mRNA levels of mesenchymal markers (fibronectin, N-cadherin, Slug) and hub genes (Il-1β, PLAUR) in U87 cells following treatment with SP600125 and TGF-β1, measured by RT-qPCR and normalized to *HPRT*. (**E**) Western blotting of N-cadherin in U87 cells after stimulation with SP600125 and TGF-β1. (**F**) Immunofluorescence of vimentin in U87 treated with SP600125 and TGF-β1. (**G**) Relative mRNA levels of pro-inflammatory cytokines IL-6 and IL-1β, and anti-inflammatory cytokine TGF-β1 in U87 cells treated with SP600125 and TGF-β1, evaluated by RT-qPCR. Data are represented as mean ± standard deviation (SD). Statistical significance was determined by comparison with control or TGF-β1-treated groups using a *t*-test. *, **, *** indicate *p*-values < 0.05, <0.01, and <0.001, respectively.

**Figure 12 ijms-26-09772-f012:**
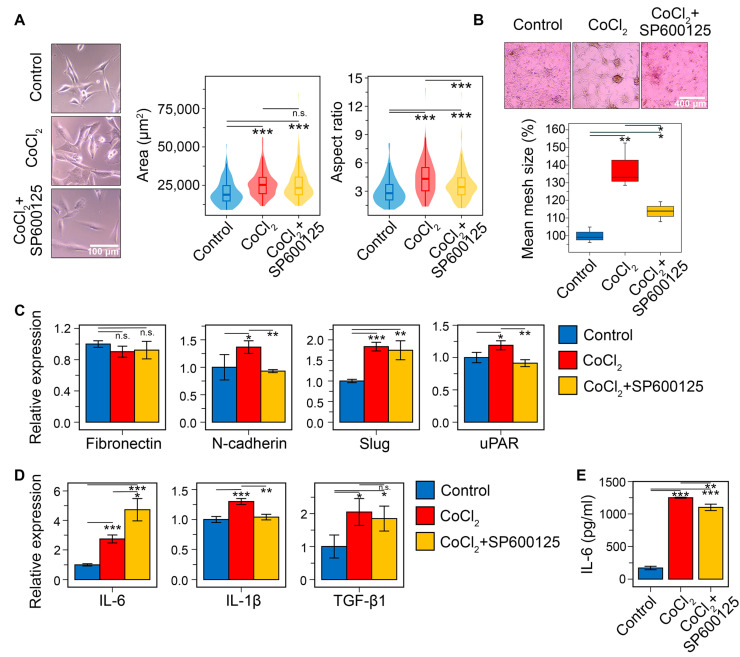
Effect of SP600125 on hypoxia-induced EMT in U87 GBM cells in vitro. (**A**) U87 cell morphology after 30 min of SP600125 (20 μM) induction and 48 h of CoCl_2_ (400 μM) stimulation. Marginal graphs in the upper and right panels show the distribution of cells by area and aspect ratio, respectively, with dashed lines indicating mean values. The image on the left was taken at ×200 magnification. (**B**) The effect of SP600125 on vasculogenic mimicry of CoCl_2_-treated U87 cells. The microscopic image was taken at ×40 magnification. (**C**) Relative mRNA levels of mesenchymal markers (fibronectin, N-cadherin, Slug, and uPAR) in U87 cells following treatment with SP600125 and CoCl_2_, measured by RT-qPCR and normalized to the housekeeping gene *HPRT*. (**D**) Relative mRNA levels of pro-inflammatory cytokines IL-6 and IL-1β, and anti-inflammatory cytokine TGF-β1 in U87 cells treated with SP600125 and TGF-β1, evaluated by RT-qPCR. (**E**) Measurement of IL-6 levels by ELISA in supernatants of U87 cells stimulated with SP600125 and CoCl_2_. Data are represented as mean ± standard deviation (SD). Statistical significance was determined by comparison with control or CoCl_2_-treated groups using a *t*-test. *, **, *** indicate *p*-values <0.05, <0.01, and <0.001, respectively.

## Data Availability

This study was carried out using publicly available data from (a) The Cancer Genome Atlas (TCGA) at https://portal.gdc.cancer.gov/projects/TCGA-GBM (accessed on 26 December 2023), (b) Gene Expression Omnibus (GEO) at https://www.ncbi.nlm.nih.gov/geo/query/acc.cgi?acc=GSE192710 (accessed on 11 January 2024), (c) Chinese Glioma Genome Atlas (CCGA) at https://www.cgga.org.cn/ (accessed on 10 September 2024), (d) Gravendeel dataset at GlioVis portal (http://gliovis.bioinfo.cnio.es/, accessed on 10 September 2024), (e) dataset of compound and genetic perturbagens at Touchstone App (https://clue.io/connectopedia/the_touchstone_dataset, accessed on 11 January 2024).

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
