# Peer review of "Transcriptomic-Driven Drug Repurposing Reveals SP600125 as a Promising Drug Candidate for the Treatment of Glial-Mesenchymal Transition in Glioblastoma"

_ijms, 2025, doi:10.3390/ijms26199772_

Round 1

Reviewer 1 Report

Comments and Suggestions for Authors

Review Report

Title: Transcriptomic-Driven Drug Repurposing Reveals SP600125 as a Promising Drug Candidate for the Treatment of Glial-Mesenchymal Transition in Glioblastoma

This is a comprehensive bioinformatics-mediated drug repurposing study aimed at discovering inhibitors of glial-mesenchymal transition (GMT) in glioblastoma multiforme (GBM). By combining transcriptomic analysis (GSVA, WGCNA, GSEA, and network reconstruction) with connectivity mapping and in vitro validation, the authors recommend SP600125 as an efficacious GMT inhibitor. The study is highly relevant to the field and addresses an urgent need for effective therapies against GBM invasiveness. The methodological approach is rigorous, and the inclusion of both computational and experimental validation increases the quality of the paper significantly.

Overall, the work is of high quality, but several areas can be improved for clarity, rigor, and effect.

Scope of Experimental Validation:

Validation is restricted to U87 cells. The inclusion of additional GBM models, particularly patient-derived lines or in vivo models, would reinforce conclusions. The effects on normal glial or neuronal cells should also be discussed to address specificity.

Mechanistic Insight:

While SP600125 is a known JNK inhibitor, its multi-target nature is acknowledged. However, the mechanistic contribution of JNK versus other targets (AhR, AURKA, MELK) is not investigated.

Presentation of Results:

The manuscript is very dense, and certain figures (e.g., network diagrams) are complex. Simplifying visualizations or providing clearer legends would improve readability. Also, the description of the workflow could benefit from a more schematic overview to guide readers through the multiple analytical steps.

Limitations and Translational Relevance:

While limitations are briefly mentioned, more critical discussion is needed on challenges related to SP600125 (low tumor specificity, potential toxicity, clinical feasibility).

The authors should consider expanding the discussion on how derivatives or alternative JNK inhibitors could overcome these issues.

Language and Style:

The manuscript is generally well-written, but some sentences are long and heavy with technical details, which may reduce accessibility. Minor language polishing is recommended.

Minor Comments

Provide further details on why GMT is different from EMT and why it is especially important in GBM.

Justify choosing SP600125 as opposed to other possible candidates like vemurafenib, which is preclinically validated.

Consider including supplementary tables with raw data from in vitro assays for transparency.

Some references on GBM invasion and EMT inhibitors could be updated to include very recent studies (2023–2024).

Recommendation-Major Revision

The manuscript presents a new and significant contribution based on sound bioinformatics and experimental data. Before acceptance, however, the authors need to:

Enrich and elucidate mechanistic insight, improve translational limitations, and increase figure clarity and text readability.

If addressed, this paper has the potential to make an important impact in the field of GBM therapeutics and drug repurposing.

Author Response

Dear Reviewer #1,

We would like to express our sincere gratitude for your thorough review and insightful comments on our manuscript. Your suggestions were instrumental in improving our study. In response to your comments, we have revised the manuscript and provide our detailed responses below. All changes in the text have been highlighted in yellow.

  1. Validation is restricted to U87 cells. The inclusion of additional GBM models, particularly patient-derived lines or in vivo models, would reinforce conclusions. The effects on normal glial or neuronal cells should also be discussed to address specificity.

Authors: Corrected. Indeed, in the initial version of the manuscript, the verification studies were performed only on the U87 cell line. We acknowledge the reviewer's valid concern regarding the potential for cell context-dependent effects of SP600125, given the high heterogeneity of glioblastoma (GBM). To evaluate the anti-GBM potency of SP600125 more comprehensively, we have supplemented our study with experiments using human U118 and murine GL261 glioblastoma cells (Fig. 10A, C, Fig. S1D). These new data demonstrate the comparable anti-GBM activity of SP600125 across all tested cell lines. Furthermore, to assess the selectivity of SP600125, we analyzed its cytotoxicity against non-transformed human fibroblasts (hFF3). We recognize that hFF3 cells are not of glial or neuronal origin; however, the use of fibroblasts still provides valuable, albeit incomplete, insight into the tumor-directed action of the compound. Considering that SP600125 has previously demonstrated a marked neuroprotective effect in murine models of MPTP-induced Parkinson's disease [1] and sepsis-associated encephalopathy [2], it is reasonable to propose that this molecule exhibits a favorable safety profile in the brain. References to these studies have also been added to the manuscript (please see p. 22, lines 713-717).

  1. While SP600125 is a known JNK inhibitor, its multi-target nature is acknowledged. However, the mechanistic contribution of JNK versus other targets (AhR, AURKA, MELK) is not investigated.

Authors: Corrected. We agree with the reviewer on the necessity of delineating the contributions of JNK1/2 and other primary protein targets of SP600125 to its anti-GBM activity. To address this, we analyzed the connectivity scores of other JNK inhibitors with the GBM signature (Fig. 9C, left heatmap). This analysis revealed that only CC-930 (a chemical derivative of SP600125 that has advanced to clinical trials) showed a negative connectivity score similar to that of SP600125, whereas all other JNK inhibitors demonstrated positive scores. These results suggest that the anti-GBM potency of SP600125 is defined by its multi-target effect. Indeed, a subsequent Connectivity Map analysis of knockdowns of SP600125's primary protein targets indicated that its GMT-targeting activity depends not only on the inhibition of JNK1 but also on interactions with MELK and FLT3 (Fig. 9C, right heatmap). These findings have been described in detail and added to the Results (please see p. 14, lines 417-431) and Discussion (p. 22, lines 693-707) sections.

  1. The manuscript is very dense, and certain figures (e.g., network diagrams) are complex. Simplifying visualizations or providing clearer legends would improve readability. Also, the description of the workflow could benefit from a more schematic overview to guide readers through the multiple analytical steps.

Authors: Corrected. In response to this comment, the following changes were made to the revised manuscript to improve the clarity of the results for the reader. First, we have clarified the workflow overview by dividing the different stages of the analysis into distinct blocks (Fig. 1) and have provided a more detailed description of each research stage in the figure legend (please see p. 4-5, lines 135-158). Second, to facilitate a better understanding of the steps involved in identifying the novel GBM signature, the corresponding workflow has been added as Fig. 3A (p. 7). Third, the multi-panel figures, which consisted of a large number of panels, have been split into more compact figures (now Fig. 3/Fig. 4, Fig. 5/Fig. 6, and Fig. 7/Fig. 8). Dear Reviewer #1, we thank you for this critical observation. We are confident that the new structure of the illustrative material will allow the reader to better understand the results of our study without becoming confused by their description.

  1. While limitations are briefly mentioned, more critical discussion is needed on challenges related to SP600125 (low tumor specificity, potential toxicity, clinical feasibility).

The authors should consider expanding the discussion on how derivatives or alternative JNK inhibitors could overcome these issues.

Authors: Corrected. We thank the Reviewer #1 for this important observation. We agree that although promising pharmacological effects have been identified, SP600125 cannot be used directly in GBM patients due to its off-target effects. A prominent example is CC-930, a known derivative of SP600125, which was effective against idiopathic pulmonary fibrosis but was withdrawn from Phase 2 clinical trials due to an increased risk of hepatic injury (NCT01203943). Nevertheless, published studies have established that SP600125 effectively blocks tumor growth in murine models [3,4] and possesses neuroprotective potential in vivo [1,2]. This indicates the molecule's ability to accumulate in tumor tissue and cross the blood-brain barrier. Therefore, SP600125 should be considered primarily as a structural platform for developing novel anti-GBM compounds. This perspective, along with potential strategies for the derivatization of SP600125, has been added to the Discussion section (please see p. 22, lines 708-728).

  1. The manuscript is generally well-written, but some sentences are long and heavy with technical details, which may reduce accessibility. Minor language polishing is recommended.

Authors: Corrected. In response to this comment, we have carefully re-read the manuscript and, indeed, identified several overly complex sentences and awkward phrases. We thank the reviewer for this valuable feedback. The text has been substantially revised as follows:

(a) Long sentences have been divided into shorter, more concise ones.

(b) Descriptions of technical details that impeded readability have been removed (e.g., p. 6, lines 178-179, line 184; p. 11, lines 327-328; p. 13, lines 383-385).

(c) In several instances, the text has been structured into itemized lists for clarity, using either numbered points (e.g., p. 13, lines 406-407; p. 19, lines 589-591; p. 21, lines 650-652) or a bulleted list (p. 21, lines 661-675).

All changes have been highlighted in yellow throughout the manuscript.

  1. Justify choosing SP600125 as opposed to other possible candidates like vemurafenib, which is preclinically validated.

Authors: Corrected. The selection of SP600125 for further study was driven by two key factors: namely, its highest centrality score in the chemical-protein network (Fig. 9A) and its superior BBB permeability score relative to vemurafenib. A detailed account of this selection process for in vitro verification has been provided in section 2.5 (please see p. 14, lines 413-416).

  1. Consider including supplementary tables with raw data from in vitro assays for transparency.

Authors: Corrected. Dear Reviewer, in response to your valuable comment, we have added references to the Supplementary Tables containing the raw data. These include the connectivity mapping of JNK inhibitors and the knockdown of SP600125's primary targets (Table S2 and S3, respectively; p. 14, lines 423 and 427), the microscopic analysis of cell morphology under GBM induction and SP600125 treatment (Tables S4-S6; p. 26, line 898), and the xCELLigence analysis of transwell migration of GBM cells (Table S7; p. 26, line 906).

  1. Some references on GBM invasion and EMT inhibitors could be updated to include very recent studies (2023–2024).

Authors: Corrected. Please see p. 2-3, lines 83-90; p. 20, lines 640-642; p.22, lines 693-694.

  1. The manuscript presents a new and significant contribution based on sound bioinformatics and experimental data. Before acceptance, however, the authors need to:

Enrich and elucidate mechanistic insight, improve translational limitations, and increase figure clarity and text readability.

Authors: Corrected. Dear Reviewer #1, we are grateful for your valuable recommendations, constructive critique, and positive assessment of our work. We have carefully considered all of your points and hope that our revised manuscript fully addresses the concerns you raised.

We hope that corrected version of the manuscript will be acceptable for publication in the International Journal of Molecular Sciences.

Sincerely,

Dr. Andrey Markov

References

  1. Wang, W.; Shi, L.; Xie, Y.; Ma, C.; Li, W.; Su, X.; Huang, S.; Chen, R.; Zhu, Z.; Mao, Z.; et al. SP600125, a new JNK inhibitor, protects dopaminergic neurons in the MPTP model of Parkinson’s disease. Neurosci. Res. 2004, 48, 195–202, doi:10.1016/j.neures.2003.10.012.
  2. Gagnani, R.; Singh, H.; Suri, M.; Bali, A. JNK inhibition mitigates sepsis-associated encephalopathy via attenuation of neuroinflammation, oxidative stress and apoptosis. Metab. Brain Dis. 2025, 40, 148, doi:10.1007/s11011-025-01563-4.
  3. Ennis, B.W.; Fultz, K.E.; Smith, K.A.; Westwick, J.K.; Zhu, D.; Boluro-Ajayi, M.; Bilter, G.K.; Stein, B. Inhibition of Tumor Growth, Angiogenesis, and Tumor Cell Proliferation by a Small Molecule Inhibitor of c-Jun N-terminal Kinase. J. Pharmacol. Exp. Ther. 2005, 313, 325–332, doi:10.1124/jpet.104.078873.
  4. Yu, H.; Wu, C.-L.; Wang, X.; Ban, Q.; Quan, C.; Liu, M.; Dong, H.; Li, J.; Kim, G.-Y.; Choi, Y.H.; et al. SP600125 enhances C-2-induced cell death by the switch from autophagy to apoptosis in bladder cancer cells. J. Exp. Clin. Cancer Res. 2019, 38, 448, doi:10.1186/s13046-019-1467-6.

Reviewer 2 Report

Comments and Suggestions for Authors

In this submission to IJMS, the authors aimed to discover potential GMT inhibitors using transcriptomic-based repurposing applied to both approved and experimental drugs. The authors used deep bioinformatic analysis of transcriptomic data from GBM patient tumors and GBM cell lines with mesenchymal phenotype. Using this gene signature, a number of small molecule compounds were predicted by the authors as potent anti-GMT agents. In vitro validation by the authors showed that SP600125, which is the most closely associated with GMT-related hub genes, effectively inhibited TGF-β1- and chemical hypoxia-induced GMT in U87 GBM cells by reducing morphological changes, migration, vasculogenic mimicry, and mesenchymal marker expression. The authors claim that their results clearly demonstrate the applicability of connectivity mapping as a powerful tool to accelerate the discovery of effective GMT-targeting therapies for GBM and significantly expand our understanding of the antitumor potential of SP600125.

I consider this work to be of interest to health researchers as well as readers of this journal. As such, I am generally supportive of publication with a few minor comments. While the authors use conventional bioinformatics approaches for their analyses, there have been recent studies using the various machine learning methods for creating mappings, which should be noted: GigaScience 2021, 10, giab071 and Environ. Sci. Technol. Lett. 2023, 10, 1017–1022. In particular, these prior studies showed that machine learning approaches such as tSNE can give more accurate mappings than the CLUE platform used by the authors. I am not asking the authors to carry out such calculations, but these recent applications of machine learning for health should also be noted.

Author Response

Dear Reviewer #2,

We appreciate your consideration and interest in our work. We have edited the manuscript taking into account your comments and provide answers to your questions.

  1. While the authors use conventional bioinformatics approaches for their analyses, there have been recent studies using the various machine learning methods for creating mappings, which should be noted: GigaScience 2021, 10, giab071 and Environ. Sci. Technol. Lett. 2023, 10, 1017–1022. In particular, these prior studies showed that machine learning approaches such as tSNE can give more accurate mappings than the CLUE platform used by the authors. I am not asking the authors to carry out such calculations, but these recent applications of machine learning for health should also be noted.

Authors: Dear Reviewer #2, we agree that machine learning is a valuable tool for computational drug repurposing. We have cited recent studies that apply machine learning to identify anti-GBM activities in known drugs (pp. 20–21, lines 638–646). Additionally, in the limitations section, we highlighted that machine learning could significantly enhance Connectivity Map analysis and have included the citations you recommended (pp. 22–23, lines 741–744).

We hope that corrected version of the manuscript will be acceptable for publication in the International Journal of Molecular Sciences.

Respectfully,

Dr. Andrey Markov

Reviewer 3 Report

Comments and Suggestions for Authors

This manuscript presents a methodologically robust and conceptually compelling study in which the researchers take on a multi-modal bioinformatic and experimental approach in search of SP600125 as a potential glial-mesenchymal transition (GMT) inhibitor in glioblastoma. Their research incorporates transcriptomic evaluation through publicly available datasets, network ranking of hub genes, connectivity mapping, cheminformatics screening, and in vitro validation within GBM cell lines. The rationale behind this work is clearly explained, and in the study, there are plans to address a serious and thus far unanswered need in the therapy of glioblastoma, particularly in targeting the mesenchymal subtype defined by therapy resistance and invasiveness. The manuscript is thorough, but a couple of crucial points require elucidation, support by experiment, and more discussion in detail. These are points involving limitation in the model, selectivity in the candidate compound, and clinical relevance in the results. That said, this manuscript does have potentially novel ideas and is worthy of consideration following extensive revision.

1. The acronym GMT is frequently utilized in the paper; however, it is comparatively less well-defined in the context of glioma biology than is the epithelial-mesenchymal transition (EMT). The paper would be helpful if a more straightforward definition of GMT was presented earlier in the Introduction and if it was made clear how it mechanistically differs from EMT, rather than asserting their analogy.

2. The final 31-gene GMT hub gene signature derivation is the focus of the study. The move, however, from the 694 Blue Module through 160 mesenchymal-up-regulated genes and on down to the final 31 Hub genes could be more succinctly explained. A table or flowchart detailing the filtering and selection process would make it clearer.

3. There is some risk of circular reasoning in the validation of the GMT signature. Since the GMT score is based on known mesenchymal and proneural gene sets, and the same datasets are used to validate hub genes, it may reinforce known biology without necessarily identifying new players. The authors should discuss how their approach avoids circularity and whether their network analysis reveals novel components not previously associated with the Verhaak signature.

4. All functional assays were performed with U87 cells. Even though it is a typical GBM model, it does not retain the primary characteristics of GBM heterogeneity and stemness. Verification in a different GBM cell line, ideally a mesenchymal characteristic or patient-derived model, would substantially enhance the reliability in generalizability of results.

5. The majority of validation of targets occurs at the mRNA level. None exists at the protein level in the case of different mesenchymal markers or N-cadherin, for different hub genes. Enlarging the techniques of Western blotting or immunostaining to involve at least a few more markers, such as PLAUR or Slug, would add biological robustness.

6. This manuscript doesn't have in vivo data or a description of how SP600125 would be in a mouse GBM model. Although in vivo evaluation is outside the range of this article, a paragraph on potential translational issues, be it pharmacokinetics, systemic toxicity, or penetrability into tumors, would be useful.

7. SP600125 is a well-known inhibitor of JNK with poor kinase selectivity and multiple off-target behaviors. The issue is briefly mentioned in the Discussion. Still, this manuscript would benefit from a greater amount of commentary on how, in future work, specificity could be corrected or whether the observed effects are completely reliant on JNK.

8. The cutoff in drug selection in the Connectivity Map analysis (below -90 summary score) is not well defended. The paper ought to describe why the cutoff was chosen and whether or not additional drugs within or close to the cutoff were considered or excluded.

9. The investigation only targets SP600125; it is thus noteworthy to indicate whether additional JNK inhibitors, e.g., JNK-IN-8, were tested in the Connectivity Map screen or a comparative study considered. In the event no such comparisons were conducted, an explanation should be provided on whether the exclusion was a function of availability barriers or otherwise.

10. The terminology used in the paper alternately intermingles the definitions of EMT and GMT. The authors need to distinguish regularly between these terms and clarify when they are referring, in particular, to glial transition processes as compared to larger mesenchymal changes.

11. The first set of bioinformatic results indicate a significant link among hub genes and immune signatures, particularly M1-like macrophages and inflammation cytokines. Still, no downstream functional study touched on whether or not SP600125 regulates these immune-associated genes or pathways. If there are any data, they need to be incorporated. If there are none, then it should be mentioned as a limitation.

12. Many Figures, especially Figures 5 and 6, include complex panels and lengthy annotations. Consider breaking them into individual figures or simplifying the figure layout in a way that helps make it more accessible to a general audience.

13. The results of statistical analysis are presented in figures by asterisks, but methods (t-test, ANOVA, etc.) are sometimes obscure in figure legends. Make sure that statistical method and thresholds of p-values are consistently presented in all panels.

14. The authors briefly confirm that SP600125 has yet to be clinically approved. However, they fail to expand on its use in animal models or if any clinical trial efforts are in place. Concise section discussing the current pharmacological status and challenges in translation would be appropriate.

Comments on the Quality of English Language

The manuscript is generally well written and communicates the scientific content effectively. The vocabulary is appropriate for a research article, and the structure is logical and coherent. However, there are occasional issues with overly long sentences, awkward phrasing, and redundancy, particularly in the Introduction and Discussion sections.

Author Response

Dear Reviewer #3,

On behalf of all the authors, I would like to express our sincere gratitude for your thorough assessment of our manuscript, your constructive criticism, your positive view of our research, and your commitment to improving our article. We have revised the manuscript according to your valuable comments and provide our point-by-point responses below.

  1. The acronym GMT is frequently utilized in the paper; however, it is comparatively less well-defined in the context of glioma biology than is the epithelial-mesenchymal transition (EMT). The paper would be helpful if a more straightforward definition of GMT was presented earlier in the Introduction and if it was made clear how it mechanistically differs from EMT, rather than asserting their analogy.

Authors: Corrected. Dear Reviewer #3, we agree that the original manuscript did not define GMT precisely. To improve clarity for a broader audience, we added a paragraph in the introduction explaining GMT and its similarities and differences compared to EMT (p.2, lines 46–63).

  1. The final 31-gene GMT hub gene signature derivation is the focus of the study. The move, however, from the 694 Blue Module through 160 mesenchymal-up-regulated genes and on down to the final 31 Hub genes could be more succinctly explained. A table or flowchart detailing the filtering and selection process would make it clearer.

Authors: Corrected. Dear Reviewer #3, thank you for highlighting these confusing points in our text. To address them, we have added a flow chart to Fig. 3 illustrating the steps in constructing the 31-gene signature. We also provided more detailed information about this workflow in the legend for Fig. 3 to enhance clarity (p. 7, Fig. 3A, lines 225–231).

  1. There is some risk of circular reasoning in the validation of the GMT signature. Since the GMT score is based on known mesenchymal and proneural gene sets, and the same datasets are used to validate hub genes, it may reinforce known biology without necessarily identifying new players. The authors should discuss how their approach avoids circularity and whether their network analysis reveals novel components not previously associated with the Verhaak signature.

Authors: Corrected. Dear Reviewer #3, thank you for your valuable comments. As the first step of our study, we used WGCNA to identify modules of co-expressed genes and found the “blue” module, which expression correlated with the GMT score. The “blue” module included 100 genes from Verhaak’s mesenchymal signature plus 594 additional genes (p. 6, lines 201-203). Next, we identified 31 hub genes, 25 of which were not part of Verhaak’s signatures (p. 7, lines 221-223). Literature text mining revealed that some of these hub genes are poorly studied in the context of GMT (Fig. 5B). Based on these results, our analysis identified new potential regulators rather than previously known players. However, six hub genes (CASP4, DAB2, THBD, PLAUR, IL4R, and TLR2) overlapped with Verhaak’s mesenchymal signature, and we highlighted them in orange color in Figs.4–6. Nevertheless, we want to emphasize that although these genes were included in Verhaak’s mesenchymal signature, not all of them have been studied in the context of GMT (p. 20, lines 622–630). We have revised the formulation in the aforementioned lines to convey these ideas more clearly.

  1. All functional assays were performed with U87 cells. Even though it is a typical GBM model, it does not retain the primary characteristics of GBM heterogeneity and stemness. Verification in a different GBM cell line, ideally a mesenchymal characteristic or patient-derived model, would substantially enhance the reliability in generalizability of results.

Authors: Corrected. Dear Reviewer #3, thank you for your methodological advice. Although primary cell lines were unavailable, we conducted additional experiments using immortalized GBM cell lines, including human U118 and mouse GL261:

  1. MTT assay on GL261 cells (p. 15, Fig. 10A, lines 445-448).
  2. Wound healing assays on U118 and GL251 cells (pp. 15–16, Fig. 10C, lines 470–471, Supplementary Figs. S1B, S1C).
  3. Morphology examination on U118 cells (p. 16, lines 484–486, Supplementary Fig. S1D).
  4. The majority of validation of targets occurs at the mRNA level. None exists at the protein level in the case of different mesenchymal markers or N-cadherin, for different hub genes. Enlarging the techniques of Western blotting or immunostaining to involve at least a few more markers, such as PLAUR or Slug, would add biological robustness.

Authors: Corrected. To strengthen validation of GMT-associated changes at the protein level, we examined vimentin expression in U87 cells treated with TGF-β1 and SP600125 using immunofluorescence (pp. 16-17, Fig. 11F, lines 509–510). To strengthen the robustness of our findings, we also verified the overproduction of the pro-inflammatory cytokine IL-6, which is associated with hypoxia-induced GBM, in U87 cells using ELISA (p. 18, Fig. 12E).

  1. This manuscript doesn't have in vivo data or a description of how SP600125 would be in a mouse GBM model. Although in vivo evaluation is outside the range of this article, a paragraph on potential translational issues, be it pharmacokinetics, systemic toxicity, or penetrability into tumors, would be useful.

Authors: Corrected. Dear Reviewer #3, thank you for this important observation. We acknowledge this oversight and agree that establishing the translational potential of SP600125 is vital for assessing its relevance to GBM therapy. In response to your comment, we have added references to studies that have previously established the significant anti-tumor activity of intraperitoneal SP600125 injections in various murine models, including human xenografts [1,2]. This confirms its ability to accumulate in tumor tissue. Furthermore, SP600125 has demonstrated significant neuroprotective activity in murine models of Parkinson's disease [3] and encephalopathy [4], indicating its capacity to cross the blood-brain barrier. This new discussion can be found on p. 22, lines 708-717.

  1. SP600125 is a well-known inhibitor of JNK with poor kinase selectivity and multiple off-target behaviors. The issue is briefly mentioned in the Discussion. Still, this manuscript would benefit from a greater amount of commentary on how, in future work, specificity could be corrected or whether the observed effects are completely reliant on JNK.

Authors: Corrected. Indeed, SP600125 is a JNK1/2 inhibitor with low specificity and the ability to interact with a broad range of protein targets. In response to your valuable comment, we conducted an additional analysis to determine whether the anti-GBM potency of SP600125 is linked specifically to its JNK1/2 inhibition or is instead defined by the multi-target nature of this molecule.

To address this, we analyzed the connectivity scores of other JNK inhibitors with the GBM signature (Fig. 9C, left heatmap). This analysis revealed that only CC-930—a chemical derivative of SP600125 that has advanced to clinical trials—showed a negative connectivity score similar to that of SP600125, whereas all other JNK inhibitors demonstrated positive scores. These results indicate that the anti-GBM potency of SP600125 is defined precisely by its multi-target effect.

Subsequently, a Connectivity Map analysis of knockdowns of SP600125's primary protein targets confirmed that its GMT-targeting activity depends not only on the inhibition of JNK1 but also on interactions with MELK and FLT3 (Fig. 9C, right heatmap). These findings have been described in detail and added to the Results (please see p. 14, lines 417-431) and Discussion (p. 22, lines 693-707) sections.

  1. The cutoff in drug selection in the Connectivity Map analysis (below -90 summary score) is not well defended. The paper ought to describe why the cutoff was chosen and whether or not additional drugs within or close to the cutoff were considered or excluded.

Authors: Corrected. The cutoff ≤-90 was chosen because it represents a strong dissimilarity, as described in Connectopedia, the official Connectivity Map manual (please, see https://clue.io/connectopedia/connectivity_scores). This explanation has been added to the methods section (please, see pp.24–25, lines 830–832).

  1. The investigation only targets SP600125; it is thus noteworthy to indicate whether additional JNK inhibitors, e.g., JNK-IN-8, were tested in the Connectivity Map screen or a comparative study considered. In the event no such comparisons were conducted, an explanation should be provided on whether the exclusion was a function of availability barriers or otherwise.

Authors: Corrected. Dear Reviewer #3, thank you for your valuable comments. We added information about other JNK inhibitors tested in the Connectivity Map screen (p. 14, Fig. 9C, lines 419–424). Among them, CC-930 showed a connectivity score similar to SP600125 but was excluded from further analysis due to predicted poor blood-brain barrier permeability (Table S1). Other JNK inhibitors were excluded because they had positive connectivity scores.

  1. The terminology used in the paper alternately intermingles the definitions of EMT and GMT. The authors need to distinguish regularly between these terms and clarify when they are referring, in particular, to glial transition processes as compared to larger mesenchymal changes.

Authors: Corrected. Dear Reviewer #3, in response to your first question, we clarified the distinction between GMT and EMT (p. 2, lines 46–63). Throughout the text, we use GMT to refer to the mesenchymal transition in GBM, EMT for the mesenchymal transition in tumors of epithelial origin, and “mesenchymal transition” to describe the general process regardless of tumor origin.

  1. The first set of bioinformatic results indicate a significant link among hub genes and immune signatures, particularly M1-like macrophages and inflammation cytokines. Still, no downstream functional study touched on whether or not SP600125 regulates these immune-associated genes or pathways. If there are any data, they need to be incorporated. If there are none, then it should be mentioned as a limitation.

Authors: Corrected. Dear Reviewer #3, thank you for pointing out this omission. To investigate how SP60015 regulates immune-associated genes, we evaluated its effect on the expression of cytokines IL-6, IL-1β, and TGF-β1 in U87 cells treated with TGF-β1 (pp. 17–18, Fig. 11G, lines 529–537) or CoCl2 (pp. 18–19, Fig. 12D, lines 569–573, 580–582). To validate these results at the protein level, we measured IL-6 secretion in CoCl2/SP600125-treated cells using ELISA (pp. 18–19, Fig. 12E, lines 573–580). The discussion of these results has also been added (p. 21, lines 685–692).

  1. Many Figures, especially Figures 5 and 6, include complex panels and lengthy annotations. Consider breaking them into individual figures or simplifying the figure layout in a way that helps make it more accessible to a general audience.

Authors: Corrected. The original Figures 3, 4, and 5 have each been split into two parts (pp. 7–10, 12–13, Figs. 3–8).

  1. The results of statistical analysis are presented in figures by asterisks, but methods (t-test, ANOVA, etc.) are sometimes obscure in figure legends. Make sure that statistical method and thresholds of p-values are consistently presented in all panels.

Authors: Corrected. The name for the used statistical method and thresholds for p-values have been added to the descriptions of all relevant figures (p. 6, lines 168–169, 173, 176-177; pp. 8–9, lines 257, 259, 261–263; p. 10, lines 285, 287; p. 11, line 316; p. 15, lines 459–460; pp. 17–18, lines 526–528; p. 19, lines 556–558).

  1. The authors briefly confirm that SP600125 has yet to be clinically approved. However, they fail to expand on its use in animal models or if any clinical trial efforts are in place. Concise section discussing the current pharmacological status and challenges in translation would be appropriate.

Authors: Corrected. Dear Reviewer #3, in response to your valuable comment, a paragraph discussing the successes and setbacks in the clinical trials of SP600125, as well as potential strategies for its derivatization to mitigate adverse effects, has been added to the Discussion section (please see p. 22, lines 718-728).

However, there are occasional issues with overly long sentences, awkward phrasing, and redundancy, particularly in the Introduction and Discussion sections.

Authors: Corrected. We thank the Reviewer #3 for this feedback. Following this comment, we performed a thorough revision of the manuscript to improve clarity. Key changes include: (a) breaking down lengthy sentences into more concise statements; (b) removing technical details that hindered readability (e.g., p. 6, lines 178-179); and (c) introducing itemized and bulleted lists to structure complex information (e.g., p. 13, lines 406-407). Furthermore, the English language has been carefully polished throughout the manuscript. All modifications have been marked in yellow.

We hope that corrected version of the manuscript will be acceptable for publication in the International Journal of Molecular Sciences.

Thank you very much!

Sincerely,

Dr. Andrey Markov

References

  1. Ennis, B.W.; Fultz, K.E.; Smith, K.A.; Westwick, J.K.; Zhu, D.; Boluro-Ajayi, M.; Bilter, G.K.; Stein, B. Inhibition of Tumor Growth, Angiogenesis, and Tumor Cell Proliferation by a Small Molecule Inhibitor of c-Jun N-terminal Kinase. J. Pharmacol. Exp. Ther. 2005, 313, 325–332, doi:10.1124/jpet.104.078873.
  2. Yu, H.; Wu, C.-L.; Wang, X.; Ban, Q.; Quan, C.; Liu, M.; Dong, H.; Li, J.; Kim, G.-Y.; Choi, Y.H.; et al. SP600125 enhances C-2-induced cell death by the switch from autophagy to apoptosis in bladder cancer cells. J. Exp. Clin. Cancer Res. 2019, 38, 448, doi:10.1186/s13046-019-1467-6.
  3. Wang, W.; Shi, L.; Xie, Y.; Ma, C.; Li, W.; Su, X.; Huang, S.; Chen, R.; Zhu, Z.; Mao, Z.; et al. SP600125, a new JNK inhibitor, protects dopaminergic neurons in the MPTP model of Parkinson’s disease. Neurosci. Res. 2004, 48, 195–202, doi:10.1016/j.neures.2003.10.012.
  4. Gagnani, R.; Singh, H.; Suri, M.; Bali, A. JNK inhibition mitigates sepsis-associated encephalopathy via attenuation of neuroinflammation, oxidative stress and apoptosis. Metab. Brain Dis. 2025, 40, 148, doi:10.1007/s11011-025-01563-4.

Round 2

Reviewer 1 Report

Comments and Suggestions for Authors

The authors successfully addressed all my concerns.

Reviewer 2 Report

Comments and Suggestions for Authors

The authors have taken into account my comments; I support publication in its current form.

Reviewer 3 Report

Comments and Suggestions for Authors

The revised manuscript includes an intensive transcriptomic-informed protocol for the identification of the GMT-targeting chemicals for glioblastoma, especially spotlighting SP600125 as an exceptional candidate. All the former reviewer comments had been fully addressed by the authors, who significantly enriched the clarity, methodology soundness, as well as translation potential for their study.